# Parameter Calibration in Global Soil Carbon Models Using Surrogate-based Optimization

Haoyu Xu[1], Tao Zhang[1, 2], Yiqi Luo[2, 3], Xin Huang[1, 2], Wei Xue[1, 2]

[1]Department of Computer Science and Technology, Tsinghua University, Beijing 100084

[2]Department of Earth System Science, Ministry of Education Key Laboratory for Earth System Modelling, Tsinghua University, Beijing 100084
[3]Center for Ecosystem Science and Society, Northern Arizona University, USA

*Correspondence to*: Wei Xue (xuewei@tsinghua.edu.cn)

**Abstract.** Soil organic carbon (SOC) has a significant effect on carbon emission and climate change. However, the current

SOC prediction accuracy of most models is very low. Most evaluation studies indicate that the prediction error mainly comes from parameter uncertainties, which can be improved by parameter calibration. The data assimilation technique has been successfully employed for the parameter calibration of SOC models. However, data assimilation algorithms, such as sampling-based Bayesian Markov Chain Monte Carlo (MCMC), generally have high computation costs and are not appropriate for complex global land models. This study proposes a new parameter calibration method based on surrogate optimization

techniques to improve the prediction accuracy of SOC. Experiments on three types of soil carbon cycle models, including Community Land Model with Carnegie-Ames-Stanford Approach biogeochemistry sub-model (CLM-CASA') and two microbial models show that the surrogate-based optimization method is effective and efficient in terms of both accuracy and cost. Compared to the predictions using the tuned parameter values through Bayesian MCMC, the root mean squared errors (RMSEs) between the predictions using the calibrated parameter values with surrogate-base optimization and the observations

could be reduced up to 12% for different SOC models. Meanwhile, the corresponding computation cost is lower than other global optimization algorithms.

## 1 Introduction

Soil organic carbon (SOC) is the largest pool of global land carbon (Todd Brown et al., 2013; Luo et al., 2015). The emission of $CO_2$, the most important greenhouse gas, from land ecosystems greatly depends on the amount of carbon stored in soils.

Moreover, anthropogenic $CO_2$ emission leads to climate warming (Houghton et al., 2001), which further stimulates soil carbon release, forming a positive feedback between the carbon cycle and climate warming (Melillo et al., 2003; Friedingstein et al.,

2006; Luo, 2007). In the fifth Coupled Model Intercomparison Project (CMIP5), the outputs of 11 Earth system models (ESMs) show great uncertainty in the SOC predictions. Despite the similarity in model structures (Huang et al., 2017), simulated soil carbon content varies six-fold, ranging from 510 to 3040 PgC, among the models (Todd-Brown et al., 2013). Only half of 11 models have a predicted global total SOC falling within the estimated range of the Harmonized World Soil Database (HWSD).

Modelled SOC is hardly corrected with the observation (Luo et al., 2015).

Considering the high similarity in the structures of the 11 ESMs, the difference in the SOC simulations mainly results from parameterizations (Todd Brown et al., 2013). Thus, parameter calibration is among the top priorities to improve prediction of global land carbon cycle dynamics (Luo et al., 2016). However, the parameter calibration with global observations has not been widely applied, owing to the high computational cost. A matrix approach has been recently developed to reorganize the

10 carbon balance equations in the original ESMs into one matrix equation without changing any modeled C cycle processes and mechanisms (Luo et al. 2003, 2017; Huang et al. 2018). The matrix land carbon cycle models can be semi-analytically solved to obtain steady-state solutions faster than the original models by tens and hundreds times (Xia et al. 2013). As a consequence, the matrix approach makes parameter estimation and calibration possible. The matrix approach has been successfully used for the parameter calibration to constrain SOC turnover and microbial process with Bayesian Markov chain Monte Carlo (MCMC)

algorithm (Harauk et al., 2014, 2015; Shi et al. 2018).

Bayesian MCMC is a sampling-based approach and usually requires a large number of simulations for building an acceptable parameter chain. For instance, over 500,000 simulations are required during the parameter calibration of soil carbon models (Xu et al. 2006). Even using high-performance computers, complex land models, like the latest version of Community Land Model (CLM5.0), require a very long spin-up time for carbon cycle simulation, leading to several hours or days for one

simulation. Although the matrix approach has been developed to enable data assimilation of global land carbon cycle models (Harauk et al., 2014 and 2015, Shi et al. 2018), Bayesian MCMC computationally is still very expensive for calibrating global land models. More effective and efficient parameter calibration algorithms are urgently needed.

The parameter calibration of SOC models can be formulated as an optimization problem that aims to minimize the output of a cost function. This cost function evaluates the difference between the outputs of model simulation and the corresponding

observations and returns a single value (e.g., RMSE) to represent the model error. Global optimization algorithms are introduced to find the minimum value of the non-linear, non-convex, and black-box problems (Hapuarachchi et al., 2001; Ma et al., 2006; Rocha, 2008). Unfortunately, the number of required simulations of most global optimizations is very large.

To reduce the number of simulations and decrease the computational cost, we, for the first time, present a surrogate-based optimization method (SBO) for calibrating the soil carbon models. Surrogate models serve as computationally inexpensive

approximations of expensive simulation models (Booker et al., 1999), such as complex geoscientific models. During the optimization process, the surrogate model can be used to determine the new promising point in the parameter space at which the expensive simulation model originally has to be evaluated. With the help of the surrogate model, many unnecessary

simulations with bad parameter values, which lead to high prediction errors, are avoided. SBO has been showed to find the near-optimal parameter values within only a few hundred simulations for different problems (Aleman et al., 2009; Giunta et al., 1997; Regis, 2011; Simpson et al., 2001).

Most studies on both global and surrogate optimizations focus on the mathematical function benchmarks like Comparing Continuous Optimizers, abbreviated as COCO (Hansen et al., 2010; Wang and Duan, 2014). However, the optimization of the mathematical functions may be extremely different from the parameter calibration of complex real-world models. In this paper, we explore the state-of-the-art surrogate optimization method for parameter calibration of three SOC models: CLM coupled with Carnegie-Ames-Stanford Approach biogeochemistry model (CLM-CASA') and two microbial models, as used in studies by Hararuk et al. (2014, 2015). Although the three models are computationally attainable for parameter calibration, we compare the performance of surrogate-based optimization to advanced global optimization algorithms and the data assimilation method to examine the potential of SBO. The SBO may be extended to other complex global land models.

In this paper, we present the structure and parameters of three SOC models in Section 2. Section 3 introduces the algorithm design of SBO. The parameter calibration results and the analysis of different parameter calibration algorithms are presented in Section 4. Section 5 discusses the calibrated results using SBO. Finally, we draw conclusions in Section 6.

## 2 Global Land Carbon Models, Data, and Cost Function

Earth system models (ESMs) are a fundamental tool for simulating climate impacts on the carbon cycle at the global scale. There are many common properties among structures of different global land carbon modules of ESMs (Luo et al. 2017). Almost all models have multiple carbon pools. Carbon is transferred among these pools (Weng and Luo, 2011). In this study, we selected three SOC models, which have been previously calibrated for their parameters with Bayesian MCMC algorithm (Hararuk et al., 2015). The first model is the soil carbon component of CLM coupled with Carnegie-Ames-Stanford Approach biogeochemistry submodel (CLM-CASA') (Oleson et al., 2004, 2008). The CLM is the land model for the Community Earth System Model (CESM). The other two SOC models are microbial models, which consider nonlinear regulation of SOC dynamics with microbial biomass.

### 2.1 CLM-CASA' C-only Version Model

The CLM-CASA' is embedded in CLM3.5. The latter includes biogeophysics and biogeochemistry sub-models. CLM-CASA' inputs carbon through net primary productivity (NPP), which is partitioned to three live biomass pools (wood, leaves and fine roots) (Fig. 1a). Dead plant materials become litter and are transferred separately to four litter pools. Litter decomposition results in part of carbon released to the atmosphere as heterotrophic respiration and part of carbon being stabilized into soil carbon pools. Organic carbon in the soil pools is decomposed partially to be released as $CO_2$ via microbial respiration and

partially to be incorporated into other soil carbon pools. One of the key model outputs to indicate SOC dynamics is the total soil organic carbon content, which is the sum of carbon in soil microbial (or active), slow and passive pools (Fig.1).

The CLM-CASA' model simulates soil carbon decomposition as a first-order decay process (Oleson et al., 2004, 2008). Based on theoretical analysis, the carbon cycle of most ESMs can be summarized with one matrix equation (Luo et al. 2001, 2017; Luo and Weng, 2011; Xia et al., 2013) as.

$$\frac{dX(t)}{dt} = A\xi(t)KX(t) + BU(t) \tag{1}$$

Where $X(t)$ is the carbon content of different pools; $\frac{dX(t)}{dt}$ is the change of the carbon content; $A$ is a matrix of transfer coefficients among different pools; $\xi(t)$ and $K$ are both diagonal matrixes, representing environmental scaling factors and baseline carbon decomposition rates, respectively; $U(t)$ is NPP, the carbon influx into the whole system and $B$ represents the partitioning coefficients of the carbon influx among plant pools. The steady state solution of equation is given by Xia et al. (2012) as:

$$X_{ss} = -(A\underline{\xi}K)^{-1}\underline{BU} \tag{2}$$

Where $\underline{\xi}, \underline{B}$, and $\underline{U}$ are the long-term averages of the environmental scalars, C partitioning among the three live pools, and NPP, respectively. The steady state soil C generated by this C-only version is in agreement with that simulated by original CLM-CASA' model (Xia et al., 2012). The structural diagram of the CLM-CASA' C-only model are presented in Fig. 1a and the parameters are described in Table 1.

## 2.2 The Microbial Models

The microorganisms catalyse various processes of land carbon cycle, such as decomposition and stabilization of SOC (Kuzyakov et al., 2000; Luo et al., 2001; Peng et al., 2009). However, most conventional SOC models, such as CLM-CASA', do not explicitly represent microbial processes. Microbially explicit models usually represent SOC decomposition by considering extracellular enzyme activities rather than simple decay constants as in the CLM-CASA' and other traditional SOC models (Schimel and Weintraub, 2003). In this study, we focused on two enzyme-driven decomposition models; one has two pools (Fig. 1b) introduced by German et al. (2012) and the other has four pools (Fig. 1c) introduced by Allison et al. (2010). We call these two models the two-pool microbial model and the four-pool microbial model, respectively. C inputs for the two models are NPP and the outputs are the carbon content of each pool at a steady state.

The two-pool microbial model is described using the following equations (Hararuk et al., 2015).

$$\frac{dMIC}{dt} = CUE \times V_{max} \times MIC\frac{SOC}{K_m+SOC} - r_d \times MIC \tag{3}$$

$$\frac{dSOC}{dt} = Input_{soil} + r_d \times MIC - V_{max} \times MIC\frac{SOC}{K_m+SOC} \tag{4}$$

Where

$$CUE = CUE_{slope} \times T_s - CUE_0 \tag{5}$$

$$V_{max} = V_{max_0} \times exp(-\frac{E_a}{R \times (T_s + 273)}) \times exp(-par_{clay} \times clay) \tag{6}$$

$$Km = Km_{slope} \times T_s + Km_0 \times exp(par_{lig} \times lignin) \tag{7}$$

where *MIC* represents the microbial biomass, $V_{max}$ is the temperature adjusted rate of SOC decomposition; $K_m$ is the half‐saturation constant for substrate‐limited SOC decomposition rate; $r_d$ is the microbial death rate; *CUE* is the microbial carbon use efficiency; $Input_{soil}$ is the carbon influx to soil, a 30-year averages of soil C input produced by CLM-CASA' (Hararuk et al., 2015); $T_S$ is soil temperature; $R$ is the gas constant (8.31 J K−1 mol−1); $CUE_0$ and $CUE_{slope}$ are the baseline microbial carbon use efficiency and its dependency on temperature, respectively; $V_{max0}$ is the maximum rate of microbial carbon uptake;

$E_a$ is the activation energy of SOC decomposition; and $Km_0$ and $Km_{slope}$ are the baseline half-saturation constant and its dependency on temperature, respectively; *lignin* is lignin content; and $par_{lig}$ is a parameter to regulate the lignin‐dependent correction factor. See Table 2 for more description of those parameters.

The four-pool microbial model from Allison et al. (2010) is described as follows:

$$\frac{dMIC}{dt} = V_{maxup} \times MIC \frac{DOC}{Kmup + DOC} \times CUE - r_d \times MIC - r_{EnzProd} \times MIC \tag{8}$$

$$\frac{dDOC}{dt} = a_{lit\text{-}to\text{-}DOC} \times Input_{soil} + r_d \times MIC \times (1 - a_{MIC\text{-}to\text{-}SOC}) + V_{max} \times ENZ \frac{SOC}{Km + SO} + r_{EnzLoss} \times ENZ - V_{maxup} \times$$

$$MIC \frac{DOC}{Kmup + DOC} \tag{9}$$

$$\frac{dSOC}{dt} = a_{lit\text{-}to\text{-}SOC} \times Input_{soil} + r_d \times MIC \times a_{MIC\text{-}to\text{-}SOC} - V_{max} \times ENZ \frac{SOC}{Km + SOC} \tag{10}$$

$$\frac{dENZ}{dt} = r_{EnzProd} \times MIC - r_{EnzLoss} \times ENZ \tag{11}$$

where *ENZ* and *DOC* are enzyme and dissolved organic carbon pools, respectively; $V_{maxup}$ is the temperature adjusted rate

of *DOC* uptake by microbes; *Kmup* is a half‐saturation constant limiting microbial uptake of *DOC*; $r_{EnzProd}$ is a rate of enzyme production; $Input_{soil}$ is C transferred from litter to soil; $a_{lit\text{-}to\text{-}DOC}$ is the fraction of $Input_{soil}$ that is transferred to *DOC*; $a_{MIC\text{-}to\text{-}SOC}$ is the fraction of dead microbes transferred to soil; and $r_{EnzLoss}$ is the rate of enzyme loss. The temperature-dependent functions are:

$$CUE = CUE_{slope} \times T_s - CUE_0 \tag{12}$$

$$V_{maxup} = V_{maxup_0} \times exp(-\frac{E_{aup}}{R \times (T_s+273)}) \tag{13}$$

$$Kmup = Kmup_{slope} \times T_s + Kmup_0 \tag{14}$$

$$V_{max} = V_{max_0} \times exp(-\frac{E_a}{R \times (T_s+273)}) \times exp(-par_{clay} \times clay) \tag{15}$$

$$Km = Km_{slope} \times T_s + Km_0 \times exp(par_{lig} \times lignin) \tag{16}$$

where $V_{maxup0}$ is the maximum rate of microbial DOC uptake; $E_{aup}$ is the activation energy of DOC uptake; $Kmup_0$ and $Kmup_{slope}$ are baseline half‑saturation constants for substrate limitation of DOC uptake and its dependency on temperature, respectively. Fifteen parameters of the four-pool microbial model are also described in Table 2.

### 2.3 Data and Cost Function

Microbial models and CLM-CASA' C-only models divide the world into 64*128 grid cells and output SOC content at each
grid (Fig. 2). The observed SOC data for parameter calibration comes from the International Geosphere Biosphere Programme – Data and Information System (IGBP-DIS) dataset (Global Soil Data Task Group, 2000). The IGBP-DIS dataset includes a 1-km resolution global land carbon data set that has been widely used in many studies to evaluate and improve models (Zhou et al., 2009; Smith et al., 2013).

The goal of parameter calibration is to improve the SOC predictions to better fit the observations. Therefore, we use the root
mean squared errors (RMSEs) between the model SOC predictions and the observations at all grid cells as the cost function. This cost function can be described as the following formula:

$$r = \sqrt{\frac{1}{N}\sum_{i=1}^{N}(X_i - O_i)^2} \tag{17}$$

Where $N$ denotes the total number of grid cells, $X_i$ and $O_i$ are the SOC of model prediction and IGBP-DIS observation, respectively. To avoid overfitting and evaluate the calibrated parameters more fairly, we separate all grid cells into a training
set and a validation set. The training set is used to guide the parameter calibration process and the validation set is used to evaluate the calibrated results. Hararuk et al. (2014 and 2015) also used this method when calibrating SOC parameters with the Bayesian MCMC approach. The experiment results in Sections 3 and 4 refer to the results for the validation set.

# 3 Surrogate-Based Optimization Algorithm Design

## 3.1 Introduction to the Surrogate-Based Optimization Algorithm

The parameters of most soil carbon models and land models have been traditionally are tuned manually (Luo et al. 2001, 2016). The manual tuning method might be effective for simple models but still highly depends on expert experience. Complex models may consist of various components from different disciplines and have hundreds or thousands of parameters. It becomes impractical for manual tuning.

Available are different parameter calibration algorithms, which have been developed based on optimization theory. The gradient search algorithms like the quasi-Newton method are introduced to search for a set of parameters with better performance in the parameter domain. These algorithms are usually efficient and fast. However, the gradient search algorithms are designed for finding the local optimum. They cannot be used to solve the multimodal problems derived from complex earth system models. In addition, they are based on the gradient information, which is unavailable for most soil carbon and land models. These models are too complex to obtain the gradient information. Thus, the parameter calibration usually becomes a black-box optimization problem. Global optimization algorithms, such as genetic algorithms and particle swarm optimization algorithms, are based on parameter generation and selection strategies. They basically are still gradient independent but can be easily used for parameter calibration of complex earth system models. Global optimization algorithms are designed to find the global minimum. However, the number of samples (model runs) might be still too large to be applicable to complex models with large number of parameters (Jones et al., 1998). Moreover, complex earth system models, for example CLM, require several hours over hundreds of cores for only one sample run and pose a special challenge for the feasibility of automatic parameter calibration.

SBO is an efficient and effective automatic parameter calibration framework. It fits a surrogate model (or response surface) based on the previous samples and uses this surrogate model to emulate the output behaviours of original models with an acceptable level of accuracy. The main advantage of SBO is to save computational costs during the global optimization by using the surrogate model instead of the original model. And the surrogate model can be continuously improved by exploiting new sample runs with the original model. With the surrogate model, the algorithm can make full use of previous samples' information and reduce the sample size, time-to-solution, as well as the computation cost. SBO has been successfully used to solve the parameter calibration of computationally expensive black-box problems (Vu et al., 2016).

## 3.2 Key Components of the Surrogate-Based Optimization Algorithm

The flowchart of the SBO is presented in Fig. 3.

First, initial sets of parameter values are generated using a sampling method. These sets are then used as inputs to run the real simulation model. Second, a surrogate model is constructed by fitting the outputs of these sample runs. The surrogate

model serves as a computationally inexpensive approximation of the expensive simulation model (Booker et al., 1999). Then in each iteration, new sample points simulated by the real model are generated according to a specific strategy. This strategy can make use of the information gained from the surrogate model and only exploits the avoidable real model runs to meet the accuracy requirement. The new sample points and their simulation outputs are used to update the surrogate model at the same time. Finally, when some stop criteria (typically the maximum number of simulations allowed) are met, the algorithm returns the optimized parameter values. During the SBO process, quite a few sample runs are generated based on the evaluation of the surrogate model and most meaningless simulations with bad parameter values are avoided. As a result, the computationally expensive model is simulated at only a few selected promising parameter points, and the surrogate model will replace the real model during the calibration process. Thus, the computation cost is reduced substantially.

Different surrogate-based optimization algorithms may have different choices with respect to the following:

✧  The sampling method to generate the initial set $S_0$.

✧  The surrogate model, which predicts the output $y$ using the given data point $x$. Before prediction, some $(x, y)$ data pairs should be given to train the model and the data are called the training set.

✧  How to decide the new points at which to run the real model in each iteration.

For the initial sampling, Monte Carlo sampling and Latin Hypercube sampling (LHS) are two main sampling methods (McKay et al., 1979; Iman et al., 1981). In Monte Carlo sampling, values are sampled from a probability distribution, which is generally a uniform distribution unless we have additional knowledge about the model and the parameters. During the LHS procedure, the range of each parameter is divided into $M$ equally probable intervals. $M$ sample points are selected to cover all intervals of each parameter. Compared to the random sampling, LHS ensures that the ensemble of random numbers is representative of the real variability of the parameters. As a result, we use LHS to generate the initial set $S_0$ (Iman et al., 1981).

There are various surrogate models, such as multivariate adaptive regression splines (Mars) (Friedman, 1991), polynomial regression models (Myers and Montgomery, 1995), radial basis functions (RBFs) (Gutmann, 2001; Müller et al., 2014; Powell, 1992; Regis and Shoemaker, 2007, 2009; Wild and Shoemaker, 2013), and kriging (Davis and Lerapetritou, 2009; Forrester et al., 2008; Jones et al., 1998).

The Mars model is an extension of naïve linear models introduced by Friedman J H. (Friedman J H., 1991). The form of Mars is presented as follows:

$$\hat{f}(x) = \sum_{i=1}^{m} c_i B_i(x) \tag{18}$$

Where $\hat{f}(x)$ represents the prediction of $y$ at the point $x$, and $c_i$ is a constant coefficient to be trained. The $B_i(x)$ is the basis function which can take one of the following three forms: a constant, a hinge function like $max\ (0, x - const)$, and a product of more than one hinge function.

The RBF model is a real-valued function. The prediction at a point $x$ using the RBF model only depends on the distance between $x$ and other points in the training set, whose outputs have been already given. The distance $r = \|x, c\|$ is generally the Euclidean distance. The radial function is the function that satisfies the property $\phi(x, c) = \phi(\|x, c\|) = \phi(r)$. The prediction at point x with the RBF model is formulated as:

$$\hat{f}(x) = \sum_{i=1}^{N} w_i \, \phi(\|x, x_i\|) \tag{19}$$

Where the $x_i$ represents the point of the training set which has $N$ points in total. Many different radial functions have been introduced and some commonly-used ones are Gaussian $\phi(r) = e^{-(\varepsilon r)^2}$, multiquadric $\phi(r) = \sqrt{1 + (\varepsilon r)^2}$, and polyharmonic spline: $\phi(r) = r^k ln(r)$. In our experiments, we choose the Gaussian radial function.

Both the kriging model and the Gaussian process regression model predict the output using a Gaussian process governed by
10 prior covariance. The $x$ and $y$ should be normalized to satisfy a normalization distribution where the means is 0 and the covariance is 1 before they are used to train the kriging model. The kriging predictor can be found as follows:

$$\hat{f}(x) = \hat{\mu} + \sum_{i=1}^{n} c_i r_i(x) \tag{20}$$

Where $\hat{\mu}$ is the estimated mean of the Gaussian process, $c_i$ is a constant representing the weight and $r_i(x) = Corr(x, x^{(i)})$ is the correlation between the $x$ and the $ith$ point $x^{(i)}$ in the training set. $\hat{\mu}$ and $c_i$ can be trained with the training set.

In addition, many machine learning regression models are also introduced, such as support vector regression (Zhang et al., 2009), artificial neural network (Behzadian et al., 2009), and random forest (Breiman, 2001).

The strategies of parameter point generation are iterative algorithms that use data acquired from previous iterations to guide new parameter point generation. Most strategies convert the parameter point generation to optimization problems using an evaluation criterion (Fig. 3). There are many different generation strategies, including Minimizing an Interpolating Surface
(MIS) (Jones, 2001) and Maximizing Expected Improvement (MEI) (Schonlau et al., 1997; Picheny et al., 2013). In MIS, the minimum of the surrogate model response surface is found and treated as the new parameter point to evaluate the real simulation model and then update the surrogate model. MEI introduces the "expected improvement" criterion. This criterion estimates the uncertainty of the surrogate model and balances the exploration and exploitation. Exploration refers to searching in an unfamiliar area of the parameter space to learn about it and avoid trapping into some local optimum. Exploitation means
fast convergence in some area. Balancing the exploration and exploitation ensures SBO can find real global optimum and does not waste more simulations on the meaningless parameter sets and areas. Another parameter generation strategy is candidate point approach (CAND) (Regis and Shoemake, 2007). In the CAND strategy, the criterion for exploitation is MIS and the criterion for exploration is the distance of the candidate point to the set of sampled parameter points from previous iterations. The previous sampled points represent the explored region and we can estimate the uncertainty with the distance to the explored
region. A weighted sum of these two criteria is used to determine the new parameter point during the SBO.

### 3.3 Design of the Surrogate-Based Optimization Algorithm for Soil Carbon Models

Based on the previous introduction of SBO, the detailed procedure of SBO can be found as in the following box.

---

Step 1: Generate an initial sampling set $S_0$.

Step 2: Run the real model and calculate the output error of the parameter points of $S_0$.

Step 3: Build the surrogate model using the parameters and the outputs generated in Step 2.

Step 4: Predict the output errors of those points that do not belong to $S_0$ using the surrogate model and determine the points at which to run the real model.

Step 5: Run the real model again for the new parameter points of Step 4 and calculate the output errors of these selected points.

Step 6: Update the surrogate model with the new data from Step 5.

Step 7: Iterate through Steps 4 to 6 until the end condition has been met.

---

The SBO scheme mentioned in previous sections is a parameter calibration framework. The key components introduced in Section 3.2 must be selected when calibrating the parameters of soil carbon models. The LHS can cover the whole parameter space with a limited number of sample points while Monte Carlo sampling usually requires much a larger number of samples. Therefore, we choose the LHS as the initial sampling strategy.

As mentioned in the previous section, many kinds of surrogate-based models have been introduced and developed. The machine learning regression models do not perform as well as RBF and kriging models, according to the evaluation of similar cases (Wang et al., 2014). In this study, we use the RBF surrogate model (RBF-SBO) as our default choice because it has been showed to perform better than other surrogate models (Müller and Shoemaker, 2014). It is easy to be implemented. Our algorithm framework also includes other surrogate models, such as kriging and Mars, and can introduce others in the future.

The surrogate model are not accurate to represent complex and nonlinear models when the SBO starts. The MIS can be very efficient but easy to trap into local optima, since the strategy does not consider the uncertainty of the surrogate model and only select the optimum of the surrogate model. The MEI eliminates the disadvantage of MIS but can only be used for the kriging surrogate model because the calculation of the expected improvement requires the standard error at the parameter point and only the kriging (Gaussian Process) surrogate model can provide the standard error (Jones et al., 1995). Finally, we use the CAND strategy as the parameter generation strategy in our algorithm to balance the exploitation and exploration of uncertain region.

## 4 Parameter Calibration Experiments

### 4.1 Experiment Configuration

In this study, we select the Bayesian MCMC approach and four advanced global optimization algorithms for comparison with our proposed SBO method. Our SBO algorithm is implemented based on the toolkit "Surrogate Model Optimization Toolbox" (Müller, 2014). Three SOC models and their cost functions are introduced in Section 2. The target of parameter calibration is to find the optimal values of parameters to achieve the minimum value of the cost function (average RMSE). Moreover, we repeat the parameter calibration process of each algorithm 50 times and use the average results for algorithm evaluation. We compare the performance of algorithms in terms of both effectiveness and efficiency. The effectiveness refers to the accuracy of the calibrated results and the efficiency can be evaluated by the required simulation times of the original SOC models.

### 4.2 Various Global Optimization Algorithms and the Bayesian MCMC Approach

The Bayesian MCMC approach and four advanced global optimization algorithms: differential evolution (DE), particle swarm optimization (PSO), shuffled complex evolution (SCE-UA), and the covariance matrix adaption evolution strategy (CMA-ES), are compared with our RBF SBO.

DE (Storn and Price, 1997) and PSO (Kennedy, 1995; Shi and Eberhart, 2009) are the representative algorithms of the evolution strategy and swarm intelligence, respectively. They both have the ability to converge quickly and outperform many genetic algorithms and simulated annealing algorithms (Price and Storn, 2006; Shi and Eberhart, 2009). SCE-UA is designed for the parameter calibration of hydrologic models and has been successfully applied to various hydrology models such as the TOPMODEL, the Xinanjiang watershed model and short-term load forecasting (Hapuarachchi et al., 2001; Ma et al., 2006; Li et al., 2007). SCE-UA tries to keep both effectiveness and efficiency by combining the local (the simplex method) and global optimization methods. Despite the difference in algorithm details, DE, PSO, and SCE-UA all generate new parameter points according to some simple mathematical formulas. Unlike these three algorithms, CMA-ES creates new parameter points based on a multivariate normal distribution (Hansen and Ostermeier, 2001; Hansen and Kern, 2004). The dependencies between parameters are represented by the covariance matrix of a normal distribution. CMA-ES has been shown to be the best global optimization algorithm in the BBOB-2009 comparison study (Hansen, 2009).

The Bayesian MCMC approach is typically designed to obtain the posterior distributions of model parameters but it can also be used to calibrate parameters to reduce the prediction error. The Bayesian MCMC consists of two main parts: sampling and parameter estimation. During the sampling part, the adaptive Metropolis (AM) algorithm, a Markov chain Monte Carlo method, is used to conduct sampling from the *prior* parameter distributions and generate a parameter chain (Haario et al., 2001). The AM algorithm has two steps: the proposing step and the moving step. A new parameter set $p^{k+1}$ is generated from the previously accepted parameter set $p^k$ through a proposal distribution $q(p^{k+1}|p^k)$. In the moving step, the probability of

acceptance is calculated according to the Metropolis criterion (Xu et al. 2006). The parameter set that is not accepted is discarded. The AM algorithm repeats the proposing step until the new parameter set is accepted. The accepted new parameter set becomes the $p^{k+1}$ set of accepted parameters in the posterior parameter distribution (Marshall et al., 2004). The proposal step is usually repeated for 50,000 to one million times to generate enough accepted parameter sets for the posterior parameter distribution. The posterior distribution is used to estimate the Maximum likelihood estimator (MLE). Hararuk et al. (2014, 2015) applied the Bayesian MCMC approach to the parameter calibration problem of the CLM-CASA' C-only model and microbial models. Hararuk et al. (2014, 2015) conducted experiments in which the proposing step required 50,000 simulations for microbial models and 1,000,000 simulations for the CLM-CASA' model. We used the code from Hararuk et al. (2014, 2015) and repeated the calibration experiments. The detailed calibration results from the Bayesian MCMC approach are presented in Table 3.

## 4.3 Results and Analysis

### 4.3.1 Effectiveness and Efficiency

Fig. 4 presents the calibrated results (RMSE) of the different algorithms we applied. For each algorithm, we only perform 100 simulations to compare the effectiveness if the simulation times are limited. As the Bayesian MCMC approach requires a large number of samples to reach a stable distribution, over 500,000 simulations have been conducted for the algorithm evaluation. Clearly, the average RMSE of the RBF-SBO is the lowest (0.6 kg/m$^2$, better than the Bayesian MCMC algorithm) for the two microbial models among all the algorithms (Fig. 4b, c). For the CLM-CASA' model, our RBF-SBO algorithm is still superior to the global optimization algorithms. The Bayesian MCMC approach performs slightly better (about 0.02 kg/m$^2$) but requires many more simulations to achieve the results (Fig. 4a).

The results of RBF-SBO also indicate less variation among the 50 repeated experiments than the global optimization algorithms for the three models. For the same reason mentioned before, the Bayesian MCMC approach has less variation than our RBF-SBO algorithm for the two microbial models. For the CLM-CASA' model, our RBF-SBO is still promising to get stable results. Among the global optimization algorithms, CMA-ES shows a very significant fluctuation (Fig. 4b, c), indicating that it is unreliable when the number of simulations is small as 100. This is because the CMA-ES requires quite a few simulations on the exploration of the parameter domain and the construction of the parameter covariance matrix. Therefore, RBF SBO is the most effective and stable one when the number of simulations is limited.

Fig. 5 shows the results in terms of average validation RMSE. We don't compare the efficiency of Bayesian MCMC since it is in nature a sampling algorithm, not an optimization algorithm. The average validation RMSE of RBF SBO is lower than the four global optimization algorithms before the number of simulations increases to 600 for two microbial models and to 200 for the CLM-CASA' model, respectively. Our RBF-SBO requires fewer simulations than the global optimization

algorithms when they reach the same RMSE value and accuracy range. Thus, our RBF surrogate optimization is also the most efficient algorithm that requires the minimum simulation times, as well as computational costs. Compared to the global optimization algorithms, SBO has two main advantages. First, SBO samples an initial parameter set to build a surrogate model, and the building process is a learning process which can better understand the parameter space, thus help conduct better optimization. Second, SBO can avoid some bad parameter points ('bad' means high prediction error), which are not supposed to be evaluated with the help of the surrogate model.

Another important observation is that the difference between the results of our RBF-SBO and the global optimization algorithms decreases as the number of simulation increases (Fig. 5). Moreover, the CMA-ES outperforms the RBF-SBO when the number of simulations exceeds 200 for the CLM-CASA' model (Fig. 5a). Our SBO can build the surrogate model with relatively good accuracy quickly, which help find a near-optimal solution with lower computation cost. However, the surrogate model is only an approximation of the real model and the accuracy might be limited due to the strong nonlinearity and the high complexity of the real model. After gaining sufficient knowledge of the original model through many simulations, the excellent global optimization algorithms, such as CMA-ES, may achieve a similar performance or even outperform our SBO, which suggests that our SBO is better to use for the parameter calibration problem of cost-expensive models.

### 4.3.2 Impact of the Model Complexity

Compared to the two-pool and four-pool microbial models, the CLM-CASA' model has 13 carbon pools and 20 parameters. Despite the increasing complexity of the CLM-CASA' model, the SBO obtains better results before conducting 200 simulations of the real model (Fig. 5a). Moreover, our SBO is always the best parameter calibration method for the two-pool and four-pool microbial models before conducting 600 simulations (Fig. 5b, c). In addition, only one global optimization algorithm, CMA-ES, has better performance compared to our SBO on the CLM-CASA' model after 200 simulations. Considering the high variance of CMA-ES on two microbial models (Fig. 4b, c), our SBO is more effective and more reliable on average.

### 4.3.3 Impact of Different Types of Surrogate Models

We select the RBF as the surrogate model in the experiments because the RBF is the widely-adopted one in many SBO algorithms (Müller and Shoemaker, 2014). In this section, we also test two other typical surrogate models, kriging and Mars. The Mars model is simple and has almost no requirements for the sample quality. Mars is very quick to train and predict. Kriging, also known as Gaussian process regression, is a method of interpolation for which the interpolated values are modelled by a Gaussian process governed by prior covariance. Kriging provides the best linear unbiased prediction of the intermediate values under suitable assumptions on the priors.

Figure 6 presents the results of kriging, Mars and RBF in terms of average validation RMSE. The performance of the three surrogate models is similar. The three surrogate models all have reasonable performance in the parameter calibration of the three types of SOC models and perform better than global optimization algorithms, indicating that our SBO is robust.

## 5 Analysis of Parameter Calibration Results

### 5.1 CLM-CASA' Model

The steady state global SOC simulations (Eq. 2) using CLM-CASA' with the default and calibrated parameter values are presented in Fig. 7a and b, which are also compared to the observed SOC pools provided by the IGBP-DIS dataset. The SOC simulation results using the calibrated parameter values from the SBO matches the observation better than that with the default parameter values (Fig. 7c) with a relatively lower RMSE. By using the calibrated parameter values, the SOC simulations are significantly improved in most parts of the world, except some grid cells in the west of Canada and the east of Russia (Fig. 7a, b and c). As a result, the CLM-CASA' simulation result with the default parameter values can only explain 33% of variation in the observed soil C, whereas that with the calibrated parameter values can explain an improved ratio (42%) of variation in the observed soil C. The unexplained variation is partly due to uncertainty in observations. To further improve the model's accuracy, we need to gain a more in-depth understanding of uncertainty sources from the data, model structure, parameters, and forcing.

Figure 8 presents the frequency distributions of the 20 calibrated parameters based on MCMC and the calibrated parameter values using the proposed SBO (the blue lines in Fig. 8). Narrow posterior distributions indicate highly sensitive parameters, consistent with the conclusions of Hararuk et al. (2014) and Post et al., (2008). The calibrated parameter values of SBO are quite similar to the responding parameter values at the peaks of posterior distributions for most highly sensitive parameters, such as the temperature sensitivity of heterotrophic respiration ($Q_{10}$) and clay effect on C partitioning from slow to passive pools ($t_7$). The parameter calibration results (RMSE) of SBO and Bayesian MCMC are similar, consistent with the parameter calibration results listed in Table 3.

Some calibrated parameter values are very close to the assigned bounds of the parameters in Fig. 8, which is usually related to the correlations among parameters. Further investigation on the covariance among parameters is necessary to explain this issue. In addition, the unreasonable setting of those bounds might be another possible reason. For instance, the calibrated c(12,12) value ($1.01 \times 10^{-3}$) reaches its lower bound, indicating that passive SOC residence time almost approaches 1000 years.

As listed in Table 1, the calibrated temperature sensitivity ($Q_{10}$) decreases from 2 to 1.74. The size of soil microbial and passive pools increases due to the longer residence time of the passive pool and lower temperature sensitivity ($Q_{10}$). The size of the slow pool, on the contrary, decreases due to the increase in the decomposition rate from the slow pool or the decrease

of its residence time. Comprehensively, the size of the SOC, which is the sum of carbon capacity in passive pools, slow pools and soil microbial pools, increases and closely approximates the observation.

## 5.2 The Microbial Models

According to the calibrated RMSE and $r^2$, the SOC simulation of the two-pool and four-pool microbial models are very similar. Without the loss of generality, we only analyse the parameter calibration results of the four-pool microbial model in this section. After parameter calibration using the SBO, the global SOC produced by the four-pool microbial model is improved, especially in China, Russia, Europe, and North America (as shown in Fig. 9). Overall, the microbial models explain a higher fraction of global variability of the observed SOC data and have lower spatial RMSEs than the CLM-CASA' model (as listed in Table 3).

The microbial models achieve better SOC predictions than that of the calibrated CLM-CASA' model in terms of the prediction of the C capacity in the low-temperature regions (Russia, Europe, North America) and in the regions with small soil C inputs (Fig. 7b and 9). The SOC contents are determined by two main factors: the soil carbon inputs and the SOC residence time (Luo et al., 2003). Considering the same soil carbon inputs of the CLM-CASA' and the microbial models, the improvement is mostly induced by the differences in the SOC residence time. In all three models, the SOC residence time is essentially controlled by temperature (Xia et al. 2013). As a result, the temperature sensitivity ($Q_{10}$) contributes to the difference across the three models. The temperature sensitivity remains constant in the CLM-CASA'. However, both the microbial models calculate spatially variable $Q_{10}$ with higher values in the low-temperature regions and lower $Q_{10}$ in the high-temperature regions, which reflects the impact of the temperature on the microbial activity. In addition, the SOC residence time can also be affected by the quality of SOC inputs and is related to the microbial decomposition processes. Fresh C input stimulates the microbial dynamics growth, resulting in an increase in the old SOC decomposition rate (i.e., priming effect) (Kuzyakov et al, 2000; Fontaine et al., 2004, 2007). Therefore, the microbial models simulate lower SOC residence times than the CLM-CASA' in the regions with a high SOC input and a high SOC residence time and the regions with a low SOC input. This is due to the nonlinearity of the substrate limitation in the microbial models (Eqs. 8 and 10), as well as the dependency of the residence time in microbial dynamics. Comprehensively, the introduction of microbial dynamics makes the microbial models predict SOC better than the CLM-CASA' model.

Figure 10 presents the posterior distributions of the parameters calculated by Bayesian MCMC and the parameter values calibrated using our SBO. According to the posterior distribution, $r_d$, $CUE_{slope}$, $CUE_0$, $E_a$, $par_{lig}$, and $par_{clay}$ are the most constrained and sensitive parameters. The calibration results of the SBO are consistent with the posterior distributions of these highly sensitive parameters (Fig. 10) except $CUE_{slope}$ and $CUE_0$. $CUE_{slope}$ and $CUE_0$ are highly sensitive owing to their influence on temperature sensitivity. Due to the difference between $CUE_{slope}$ and $CUE_0$, the RMSE of SBO is 1.4 $kg/m^2$ and

0.8 $kg/m^2$ lower than those with Bayesian MCMC for four-pool and two-pool microbial models respectively (as listed in Table 3). The mismatch of $CUE_{slope}$ and $CUE_0$ may be mainly due to the different targets of the parameter selection between the two methods.

## 6 Conclusions

Parameter calibration is becoming more and more challenging for SOC model development, especially for the computationally-expensive global land models owing to the large number of simulations. In this study, we introduce an SBO algorithm to the parameter calibration of three SOC models. The main findings are:

1) Compared to advanced global optimization algorithms, SBO is more effective and more efficient on average. Our RBF SBO outperforms other parameter calibration algorithms when the number of simulations is no more than 200.

2) The parameter optimization based on RBF surrogate model gains more accurate calibration results than those of the Bayesian MCMC approach in the three soil carbon models.

3) The SBO scheme is robust. Various types of surrogate models have similar performance in the parameter calibration tasks of SOC models.

4) Although SBO is only guided by a single cost function, it can still result in better parameter values than the default ones. We carefully analyze the spatial SOC distributions produced by the models with the calibrated parameters using SBO, which indicates that SBO truly improves the model's prediction and simulation capability.

Although the three SOC models used in this analysis are not computationally unattainable for parameter calibration, what we have learned about SBO from this study can be potentially applied to more complex models. Currently, more and more complex simulation models present challenges to the SBO algorithm. To improve the accuracy of SBO, better surrogate models are expected. Current surrogate models including our implementation for soil carbon models mainly employ only one surrogate model, which may limit the successful use for different kinds of models. We will focus on the application of multiple surrogate models using ensemble learning in the future.

## 7 Code and data availability

The code and data of three models and the related algorithm implementations can be found in the supplement. If you have any problem when using the code and repeating the experiments, please feel free to contact the corresponding author of this paper: Wei Xue (xuewei@tsinghua.edu.cn).

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

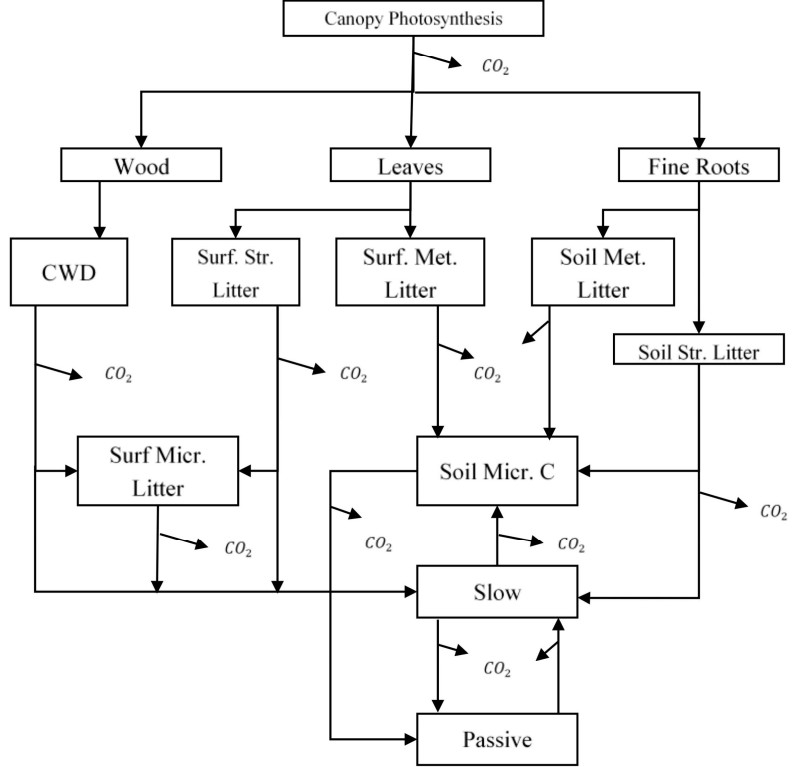

(a) **The CLM-CASA' model**

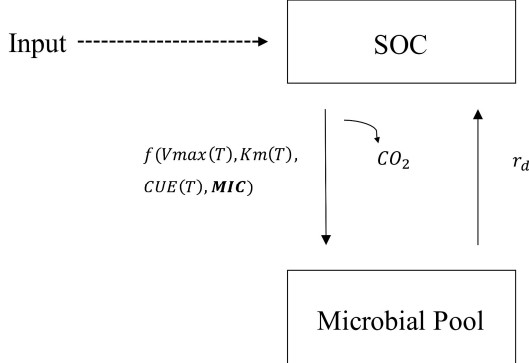

(b) **Two-pool microbial model**

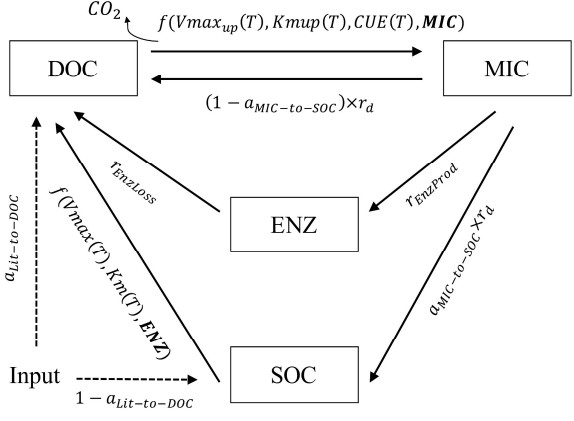

$CO_2$

$f(Vmax_{up}(T), Kmup(T), CUE(T), \boldsymbol{MIC})$

DOC

MIC

$(1 - a_{MIC-to-SOC}) \times r_d$

$r_{EnzLoss}$

$r_{EnzProd}$

ENZ

$a_{Lit-to-DOC}$

$f(Vmax(T), Km(T), \boldsymbol{ENZ})$

$a_{MIC-to-SOC} \times r_d$

Input

SOC

$1 - a_{Lit-to-DOC}$

(c) **Four-pool microbial model**

**Figure 1. Schematic representations of (a) CLM-CASA' model, (b) two-pool microbial model and (c) four-pool microbial models**

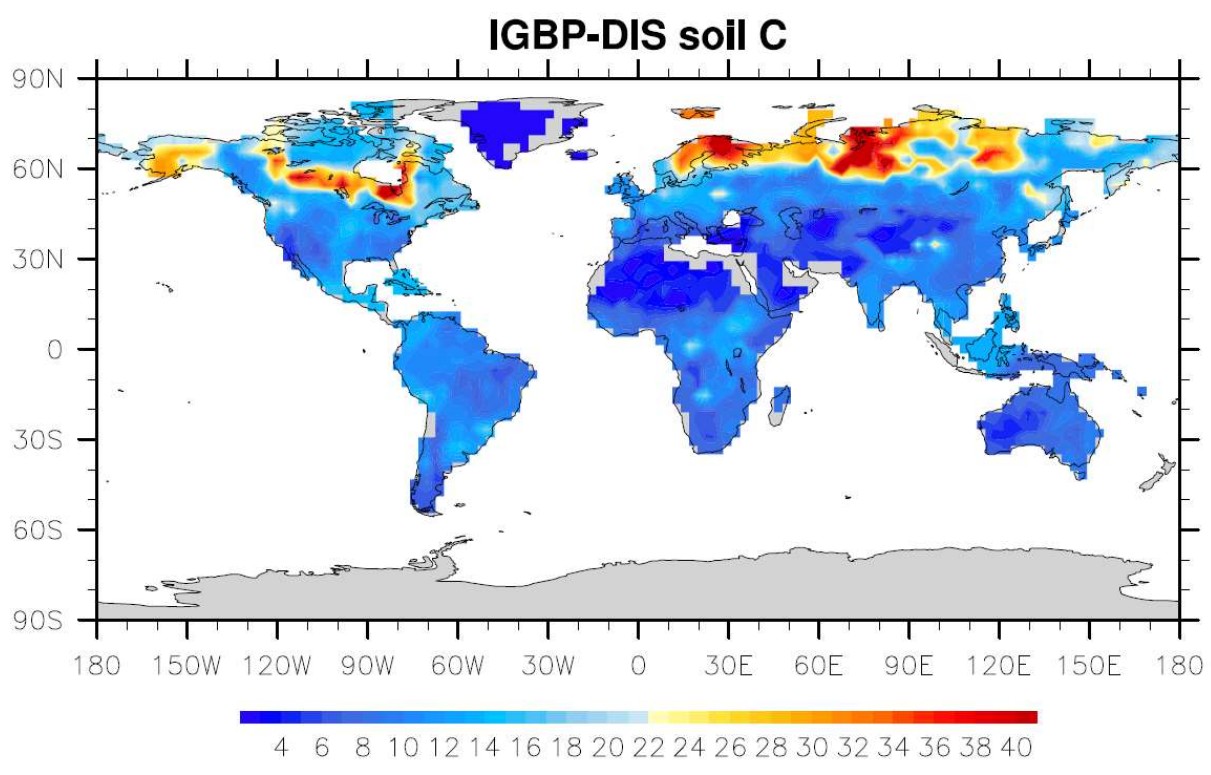

**Figure 2. IBGP-DIS soil carbon distribution. Soil carbon varies from 0 $kg/m^2$ in deserts to 60 $kg/m^2$ in boreal regions**

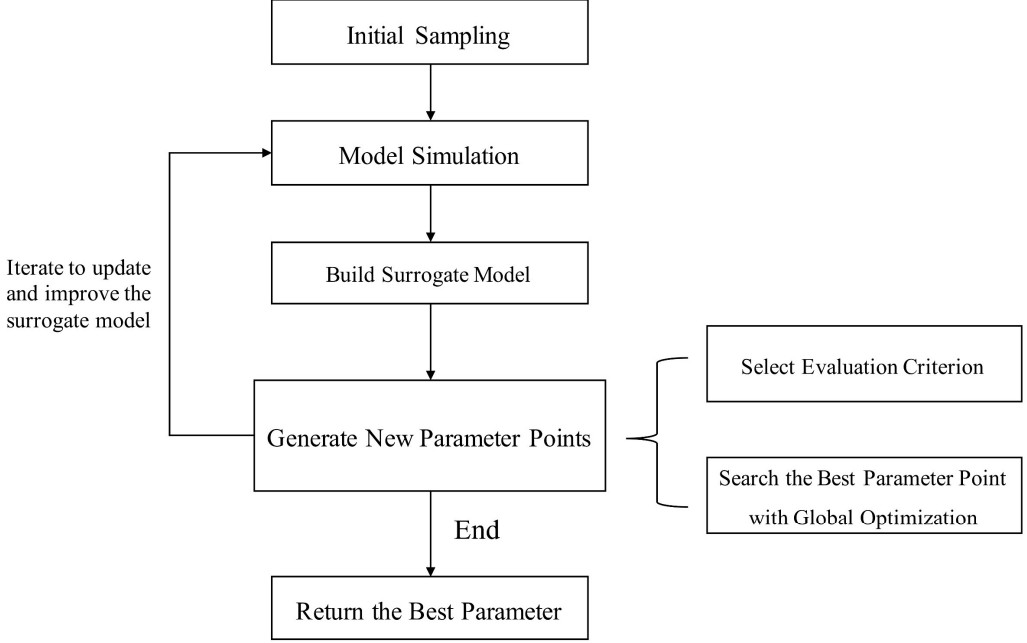

**Figure 3. The flowchart of the surrogate-based optimization**

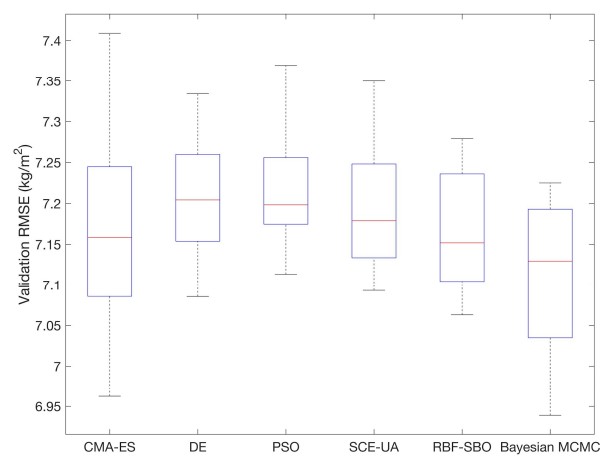

**(a) CLM-CASA' model**

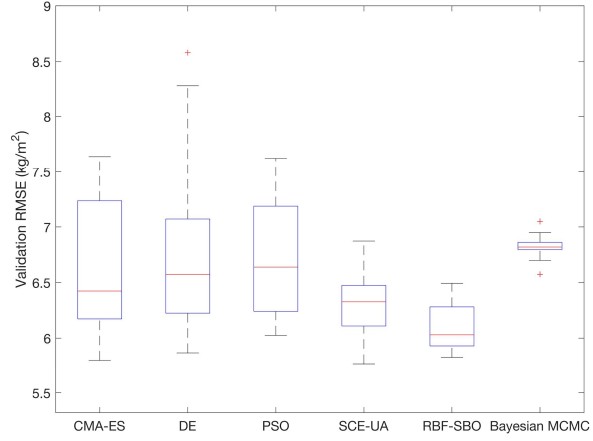

**(b) Two-pool microbial model**

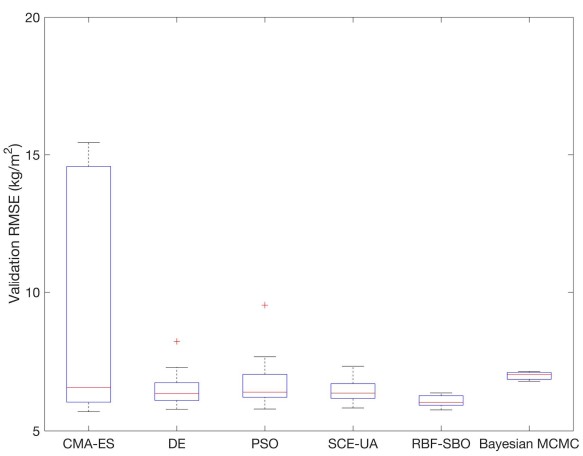

**(c) Four-pool microbial model**

**Figure 4. The RMSEs of different optimization algorithms: (a) CLM-CASA' model; (b) two-pool microbial model and (c) four-pool microbial model. The box plots show the means and the quartiles spreading over total 50 calibration runs. The central line indicates the median; the bottom and top of the box are the first and third quartiles; the black bottom and top lines out of the rectangles are the maximum and minimum; the red crosses represent the outliers. The simulation times of former 5 algorithms are 100 and the simulation times of Bayesian MCMC are presented in Table 3.**

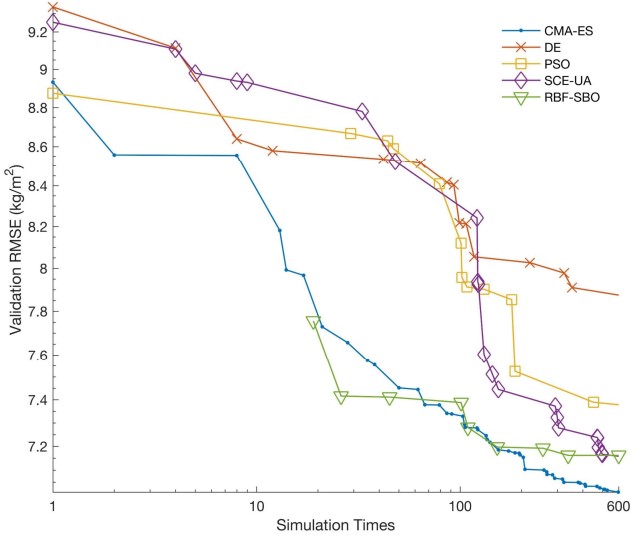

**(a) The RMSEs for CLM-CASA' model**

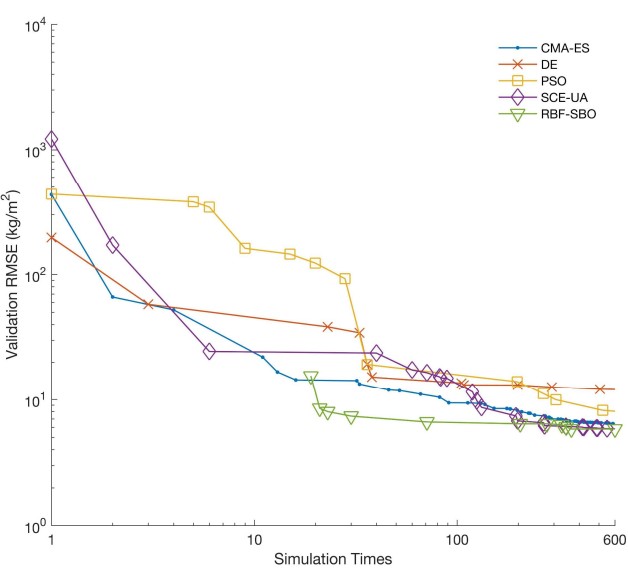

**(b) The RMSEs for two-pool microbial model**

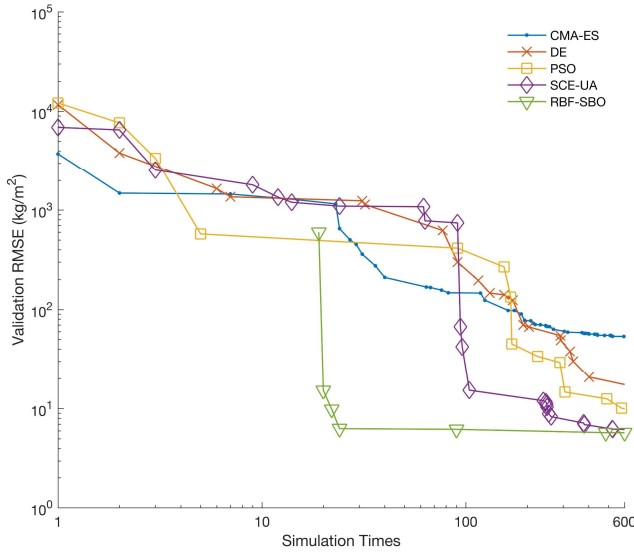

**(c)   The RMSEs for four-pool microbial model**

**Figure 5. The average RMSEs with the increase of simulation times and different optimization algorithms: (a) the CLM-CASA' model; (b) two-pool microbial model and (c) four-pool microbial model. Since RBF SBO requires some initial simulations to start optimization process, RBF SBO starts when x-axis value is 19.**

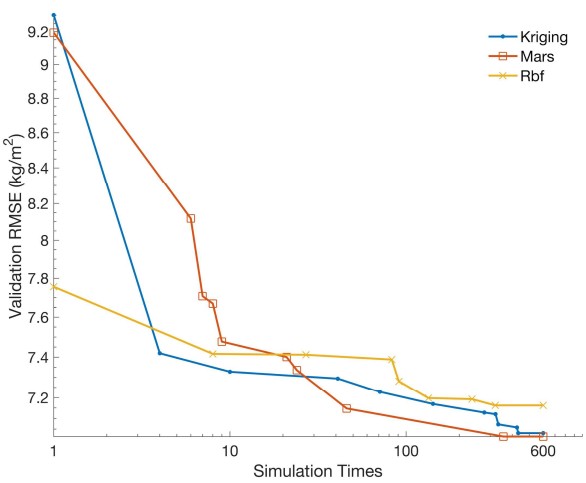

**(a)   The RMSEs for CLM-CASA' model**

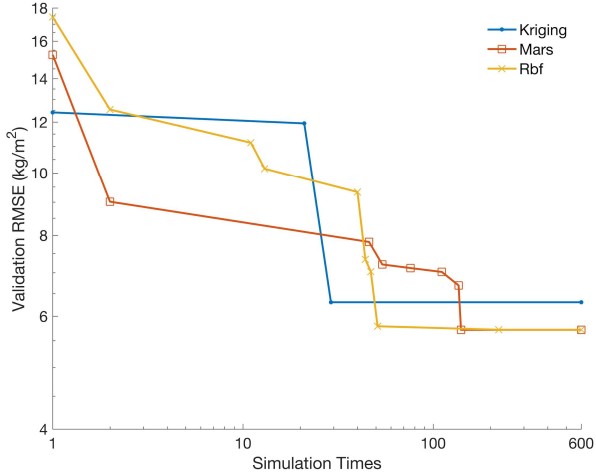

**(b) The RMSEs for two-pool microbial model**

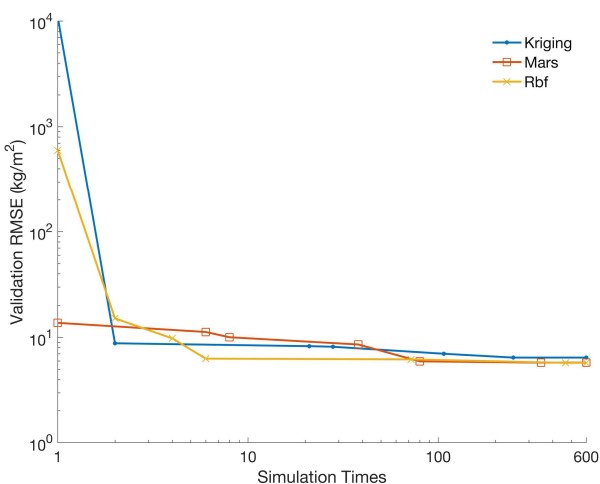

**(c) The RMSEs for four-pool microbial model**

5 **Figure 6. The average RMSEs with the increase of simulation times and different surrogate models.**

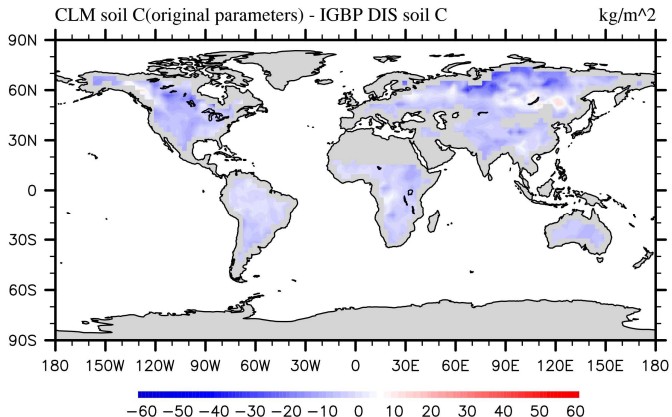

CLM soil C(original parameters) - IGBP DIS soil C     kg/m^2

**(a) CLM soil C – IGBP-DIS soil C**

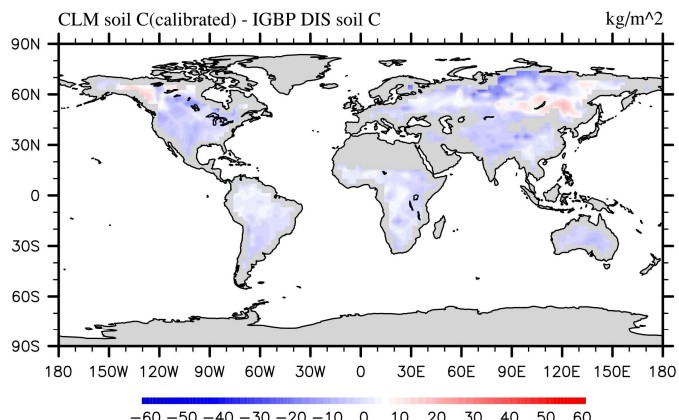

CLM soil C(calibrated) - IGBP DIS soil C     kg/m^2

**(b) CLM soil C (calibrated) – IGBP-DIS soil C**

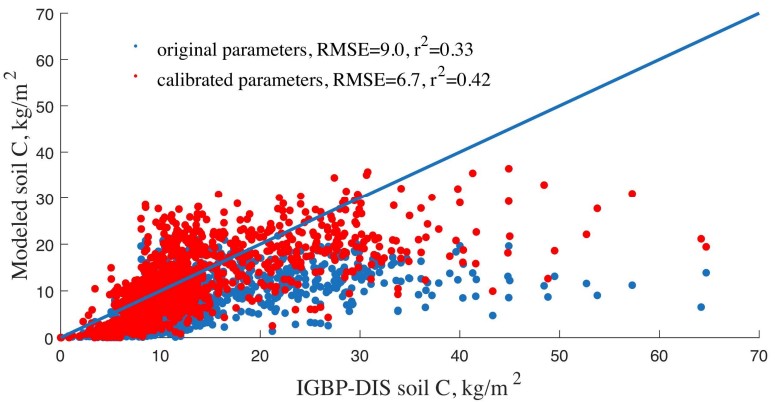

original parameters, RMSE=9.0, $r^2$=0.33

calibrated parameters, RMSE=6.7, $r^2$=0.42

**(c) Spatial correspondence between modelled soil and IGBP-DIS soil**

**Figure 7. Spatial correspondence of SOC produced by CLM-CASA' to SOC reported by IGBP-DIS. The subgraph (a) shows the results using the default parameter values and the subgraph (b) shows the results after parameter calibration using the surrogate-based optimization. The points in Fig. 7c represent the grid cell values (blue ones for the results with default parameter values and red ones for the results after parameter calibration). CLM-CASA' with the default parameter values explains 33% of variation in the observed soil C, while CLM-CASA' with the calibrated parameter values explains 42% of variability in the observed soil C.**

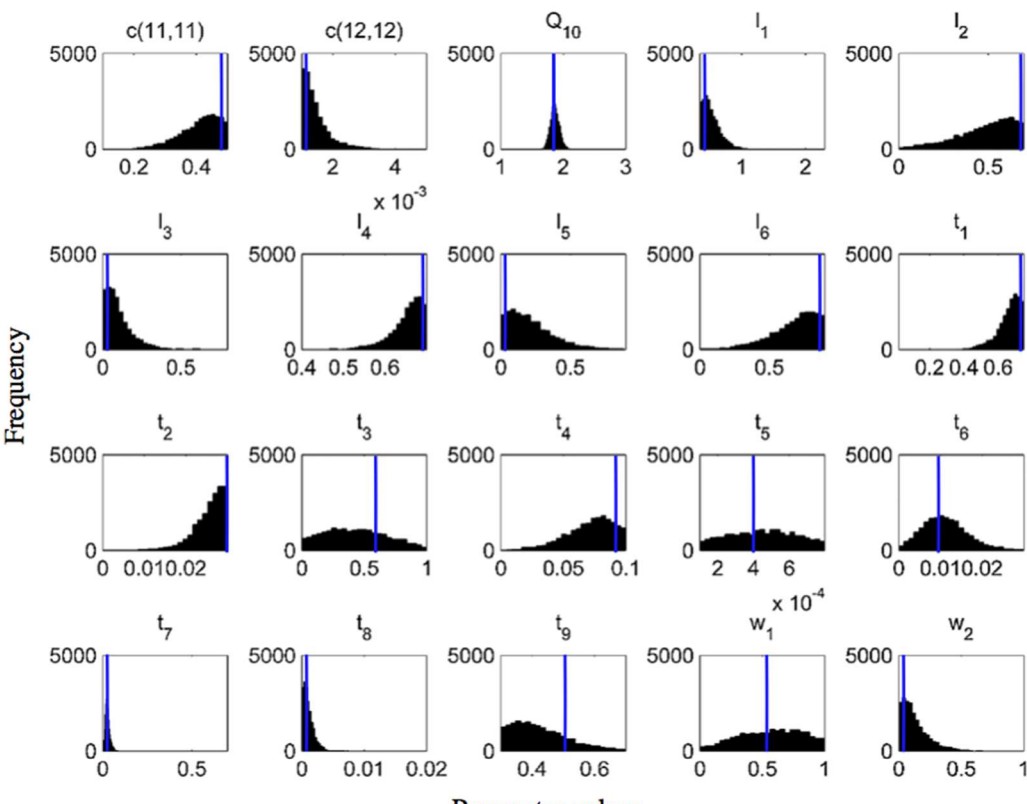

**Figure 8. Frequency distributions of 20 calibrated parameters of CLM-CASA' model by Bayesian MCMC approach (Harauk, 2014) and surrogate-based optimization (blue line in each subgraph).**

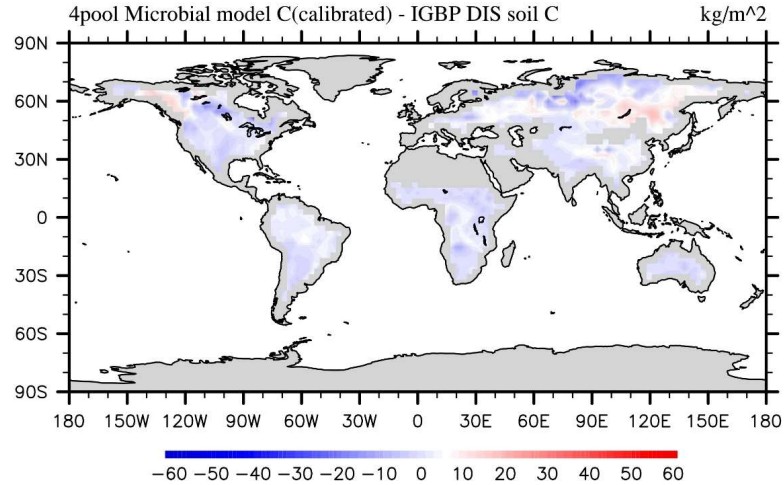

**Figure 9. Spatial correspondence of four-pool microbial model produced SOC to the IGBP-DIS reported SOC.**

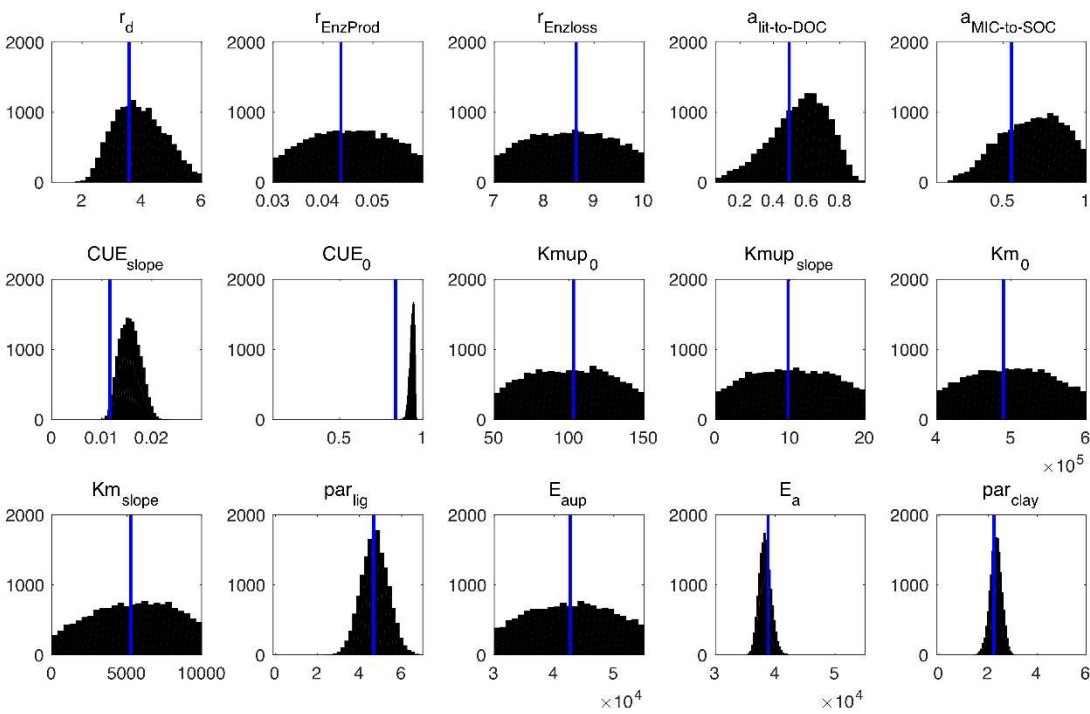

**Figure 10. Posterior probability density functions of the four-pool microbial model parameters (generated by Bayesian MCMC).**
**The blue vertical lines are the final calibrated parameter values by our surrogate-based optimization.**

**Table 1. Parameter description of CLM-CASA' C-only model**

| Parameter Description | Symbol | Default Value (x0.001) | Calibrated Value by SBO (x0.001) |
|---|---|---|---|
| Decomposition rate from slow pool | $c(11,11)$ | 200 | 495.6 |
| Decomposition rate from passive pool | $c(12,12)$ | 4.5 | 1.01 |
| Temperature sensitivity of C decomposition | $Q_{10}$ | 2000 | 1737 |
| Labile C fraction effect on C partitioning from leaves to surface metabolic litter | $w_1$ | 1000 | 589.04 |
| Labile C fraction effect on C partitioning from roots to soil metabolic litter | $w_2$ | 200 | 4.52 |
| Partitioning from surface structural to surface microbial pool if no lignin in surface structural litter | $l_1$ | 400 | 384.5 |
| Lignin effect of partitioning from surface structural litter to surface microbial litter | $l_2$ | 400 | 689 |
| Lignin effect on partitioning from surface structural litter to soil slow pool | $l_3$ | 700 | 7.499 |
| Partitioning from soil structural to soil microbial pool if no lignin in soil structural litter | $l_4$ | 450 | 697.7 |
| Lignin effect on partitioning from soil structural litter to soil microbial pool | $l_5$ | 450 | 54.46 |
| Lignin effect on partitioning from soil structural litter to soil slow pool | $l_6$ | 700 | 871.5 |
| C partitioning from soil microbial pool to slow pool if no sand or clay | $t_1$ | 169 | 747.7 |
| Clay effect on C partitioning from soil microbial pool | $t_2$ | 5.44 | 29.6 |
| Sand effect on C partitioning from soil microbial to slow pool | $t_3$ | 678 | 636.8 |
| Combined effect of sand and clay on C partitioning from soil microbial pool | $t_4$ | 22 | 99.5 |
| C partitioning from soil microbial to passive pool if no sand or clay | $t_5$ | 0.51 | 0.152 |
| Sand effect on C partitioning from soil microbial to passive pool | $t_6$ | 2.04 | 12.99 |
| Clay effect on C partitioning from slow pool to passive pool | $t_7$ | 4.05 | 24.2 |
| C partitioning from slow to passive pool if no clay | $t_8$ | 14 | 0.012 |
| C partitioning from slow to soil microbial pool if no clay | $t_9$ | 449 | 368.8 |

**Table 2. Parameter and description of the four-pool microbial models**

| Parameter Name | Parameter Description | Default Value | Calibrated Value by SBO |
|---|---|---|---|
| $r_d$ | Microbial death rate | 4.38 | 4.89 |
| $CUE_0$ | Baseline microbial carbon use efficiency | 0.63 | 0.965 |
| $CUE_{slope}$ | $CUE_0$ dependency on temperature | 0.016 | 0.00853 |
| $Km_0$ | Baseline half saturation constant | 500000 | 498467 |
| $Km_{slope}$ | $Km_0$ dependency on temperature | 5000 | 9751 |
| $E_a$ | Activation energy of SOC decomposition | 47000 | 36669 |
| $par_{clay}$ | Clay limitation | 0 | 2.41 |
| $par_{lig}$ | Lignin-dependent correction factor | 0 | 6.23 |
| $r_{EnzProd}$ | Rate of enzyme production | 0.0438 | 0.0361 |
| $r_{EnzLoss}$ | Rate of enzyme loss | 8.76 | 8.08 |
| $a_{lit-to-DOC}$ | Fraction of $Input_{soid}$ that is transferred to soil | 0.3 | 0.832 |
| $a_{MIC-to-SOC}$ | Fraction of dead microbes transferred to soil | 0.5 | 0.716 |
| $Kmup_0$ | Baseline half-saturation constants for substrate limitation of DOC uptake | 100 | 134 |
| $Kmup_{slope}$ | $Kmup_0$ dependency on temperature | 10 | 4.62 |
| $E_{aup}$ | Activation energy of DOC uptake | 47000 | 34811 |

**Table 3 Calibration results of Bayesian MCMC and our surrogate-based optimization**

| SOC model | Detail | Method | Lowest RMSE $(kg \cdot m^{-2})$ | Variance Explained | Number of Simulations |
|---|---|---|---|---|---|
| Two-pool microbial | 8 parameters 2 carbon pools | Bayesian MCMC | 6.609 | 51.6% | 50,000+500,000 |
| | | RBF-SBO | 5.785 | 51.6% | 221 |
| Four-pool microbial | 15 parameters 4 carbon pools | Bayesian MCMC | 7.142 | 51.3% | 50,000+500,000 |
| | | RBF-SBO | 5.756 | 51.4% | 199 |
| CLM-CASA' | 20 parameters 13 carbon pools | Bayesian MCMC | 7.000 | 41.0% | 50,000+1,000,000 |
| | | RBF-SBO | 7.162 | 42.8% | 321 |

