# Peer review of "Parameter Calibration in Global Soil Carbon Models Using Surrogate-based Optimization"

_Geoscientific Model Development, 2017_

## Referee Comment (RC1) · Anonymous Referee #1 · 11 Jul 2017

This manuscript describes the performance of a surrogate-based approach to calibrating three different soil carbon models relative to three global optimization algorithms and a MCMC algorithm. The results indicate that the surrogate-based optimization employing radial basis functions outperforms the other approaches in nearly all circumstances. Model calibration improves the fit of the models to a global dataset of soil carbon values, with the models incorporating soil microbial dynamics explicitly fitting the data more effectively than a model based on CLM-CASA.

Unfortunately, the quality of the English throughout the manuscript is extremely poor with numerous grammatical errors throughout all the text. Without a great deal of additional editing for language alone this will not be publishable in GMD.

All these English language errors, which are far to numerous to call out individually,

make it very difficult to undertake a review of scientific merit, but there are a number of areas that clearly require further elaboration and clarification.

Whilst it is certainly challenging, the authors assertion that it is not possible to optimize parameters directly in land surface models such as CLM is not true – see for example Post et al., 2017 JGR-B and reference there in.

The assertion that the "structures of land carbon cycle" with ESMs "are almost the same" maybe true but requires evidence and references.

It is unclear what are the differences between CLM, CLM-CASA and CLM-CASA C-only. My interpretation is that CLM-CASA C-only is the steady-state approximation detailed in Xia et al, 2012, and the SBO was developed for this. Some additional detail is required here – for example, what are the meteorological drivers, what are the inputs? "NPP" is mentioned, but never explained. This is important, as the relevance, or otherwise, of this work to informing ESM development can only be understood if the implications of using a surrogate model to parameterize a matrix-based approximation of the steady-state of the simplistic soil component of an old land model are fully articulated.

The description of how the specific SBO algorithm and parameter point generation strategies is unclear – what is about the nature of the algorithms chosen that makes them appropriate for this particularly use case?

Given the code available in the supplementary material, it is apparent that the various optimization algorithms were implemented in Matlab and relies heavily on material from the File Exchange. Details of this implementation need to be in the main text.

As the authors highlight, "sample size, the nonlinearity and complexity of the real model" all impact surrogate performance. This is partially addressed through the use of three models with different numbers of pools/parameters but not well explained, nor is there reference back to the role of surrogates with ESMs of full complexity.

The analysis of the results (Section 5) fails to discuss the implications of the optimizations for CLM-CASA C-only. What does it mean for the model if even when optimized it can only explain 40% of observed variation? Why are so many parameter values right at the edge of their prior range? Are the numbers "biological feasible"? To what extent is the improvement in fit with microbial model due to the inclusion of microbes, or rather due to spatially varying base rates?

Overall, the work described in this manuscript has the potential to inform future land surface model developments, and highlights the possibilities of using surrogate-based optimization at a fraction of the computational cost of MCMC-type approaches. With much improved editing, clarification of the points outlined here, and a more involved discussion of the outcome of optimization exercise – which is the point of the whole exercise after all – hopefully it can be considered more favorably for inclusion in GMD in the future.

---

## Referee Comment (RC2) · M. Braakhekke (Referee) · 21 Aug 2017

**Review of a "Parameter Calibration in Global Land Carbon Models Using Surrogate-based Optimization" by Xu et al.**

Maarten Braakhekke

**General comments**

This manuscript presents a novel approach for calibrating global land carbon cycle models that are computationally costly (i.e. need a long time for a single simulation). The approach, dubbed surrogate-based optimization, uses a uses a computationally cheap surrogate model, which mimics the original model, to generate candidate parameter sets at each iteration. Since the original model is only run for "good" new parameter sets, this approach avoids evaluation of the original model for "bad" parameters, thereby substantially reducing the number of model iterations, and thus computation time. The authors apply the algorithm for the CLM-CASA global land surface model in order to optimize parameters related to soil carbon cycling against global gridded soil carbon stocks from the IGBP-DIS dataset. Additionally, the approach is applied for two other soil carbon models, which explicitly represent microbial dynamics. The calibration results are compared to those of four other optimization schemes and a Bayesian MCMC algorithm.

To my mind the approach is very promising and helps to tackle an important issue with calibrating global models: the high computation cost. I'm not very experienced with optimizing global models and thus I cannot say if there are other techniques that achieve the same thing, and how these compare to the approach presented here. Nevertheless, I think the paper is relevant, and a strong contribution to the field of global modelling. Furthermore, I think the authors were quite thorough in testing the new approach by applying it to three models, and comparing the results to five other optimization/sampling schemes.

However, I do have several criticisms that should be addressed. These relate mainly to the text.

1. The fact that the RBF-SBO starts out with a considerably lower RMSE for all three models (Figure 5) suggests that the calibration setup somehow gives RBF-SBO an unfair advantage over the other algorithms. If this is the case, it would have serious consequences for the paper. Possibly the calibrations would have to be redone in a setup that removes this advantage.

2. The description of the methods need to be considerably expanded since much important information is missing, most importantly, on the algorithm itself. Ideally, one should be able to reproduce the approach from the description in the main text, appendix, or supplemental information. However, in this manuscript not nearly enough information is provided for this. For example, I would guess that the algorithm evaluates and rejects several proposal steps using the surrogate model, before a parameter set is deemed good enough to be evaluated by the true model. However, no information is provided as to how these kinds of choices are made. I would suggest including a pseudo-code block to describe the working of the algorithm. Additionally, the surrogate model is constructed based on "radial basis functions" but no additional information is given on how this works. Since the approach for surrogate model is a critical choice (as acknowledged by the authors, P6, L14) this approach needs to be described in much more detail.

There are several other places in the text where more information should be provided. These are given in the specific comments below. Parts of these descriptions may be placed in an appendix or online supplement.

3. I find the paper a bit biased towards a positive assessment of the algorithm and superiority over other algorithms. The paper would benefit from an additional discussion section on the possible limitations of the approach, which I'm sure exist. For example, the limitations of using a surrogate model for mimicking complex models is briefly mentioned (P9, L5-11), but its consequences are not further discussed. Furthermore, the SBO based estimates strongly disagree with the MCMC estimates for two of the 4-pool microbial model (CUE_slope, and CUE_0; Figure 10). This is briefly mentioned (P11, L19) but not further discussed.

4. The language in the paper is in general quite poor. There are quite a few spelling and grammar errors, and many sentences are semantically incorrect (e.g. missing or incorrect usage of articles), awkward, or use spoken rather than written English. I've listed a number of them below, but I strongly advise proof-reading by a proficient an editor proficient in the English language. Please check also the citation references, both in the text and in the bibliography. There appear to be quite a few mistakes.

5. From what I can understand from the paper (P3, L5-15) the authors only ran and calibrated soil carbon models, no full land carbon model. Therefore, I find the title somewhat misleading. The approach can probably be used to optimize a full land carbon model, but this has not been shown. I could imagine that the limitations posed by using a surrogate model would become more relevant for a full land carbon model. Hence, I would suggest replacing "land carbon models" with "soil carbon models", or "the soil carbon component of land carbon models".

**Specific comments**

**Abstract**
- P1, L11: I suggest to either replace "which can be obviously improved" with "which can obviously be improved", or remove "obviously" altogether

**Section 1**
- P1, L21: "SOC is the largest pool of global land carbon." please provide a reference for this statement.
- P1, L21: I suggest replacing "a famous" with e.g. "the most important".
- P1, L29: I suggest elucidating "agree with". E.g. "For only half of the 11 models the predicted global total SOC falls within estimated range of the HWSD"
- P1, L29: remove "s" in "coefficients"
- P2, L3: remove "the" before "parameter"
- P2, L4: "replace "expensive" with "high"
- P2, L7: remove "the" before "high"
- P2, L8: put "like CLM" between comma's. Also, CLM has not previously been introduced (it is two lines below)
- P2, L15: remove "also"
- P2, L17: replace "the" at the end of the line with "an"
- P2, L18: add "the" before "surrogate"

- P2, L23-24: "Quite a few…benchmark. This sentence is unclear. Consider revising.
- P2, L25: add a comma after "Here"
- P2, L29-30: "On average…Bayesian MCMC". This sentence presents results, and should not be in the introduction. However, I admit this may be a matter of style.
- P2, L30: It is rather unfair to compare computational cost of the SBO approach presented here to that of Bayesian MCMC, since the latter is a *sampling* algorithm, whereas the former is a *optimization* algorithm. Sampling schemes are intended to obtain a detailed approximation of the posterior/likelihood function whereas optimization schemes only yield an estimate of the maximum likelihood point. Comparing the computational cost to that of the other optimization approaches would make more sense.
- P2, L34: Replace "analysis" with "discusses"

**Section 2**
- P3, L1-2: "…their structures of land carbon cycle are almost the same". This is statement is a major oversimplification. I would suggest something like "there are many similarities"
- P3, L6: remove the "s" at the end of "carbons"
- P3, L9: "one of the most popular earth system models in the world". I suggest replacing with "widely used Earth system model"
- P3, L23: add "model" after "CLM-CASA'"
- P3, L24: add "of" after "linear"
- P3, L30: I suggest replacing "The steady solution of equation (1) is solved by Xia et al. (2012):" "the steady state solution of equation is given by (Xia et al. 2012):"
- P4, L1: add a comma after "NPP"
- Section 2.2: The microbial soil carbon models and the corresponding equations (3)-(16) need to be better explained (e.g. what processes do the different terms in the ODEs represent). For someone not experienced with such models it is currently difficult to understand what's going on.
- P4, L21: add "be" after "to"
- P5, L12: add "The" to the start of the sentence and remove the "s" in the second "models"
- Figure 2: Why do gridcells near coastlines have no data?
- P5, L22: replace "gird" with "grid"
- Tables 1 and 2: please provide the units of the parameters

**Section 3**
- Section 3: as discussed above the radial basis functions approach needs to be explained, as well as the approach to generate proposal samples
- P6, L9: I suggest adding "surrogate" before "model"
- P6, L12: I suggest replacing "cancelled" with "avoided"
- P6, L133: I suggest replacing "save much" with e.g. "substantially reduce"
- P6, 24-26: This sentence is rather vague. What is meant with "real variability"?
- P6, L24: please provide a reference for Latin hypercube sampling
- P6, L24: add "which" between "for" and "LHS"
- P7, L3: I suggest replacing "optimum" with "optima"
- P7, L7: I suggest replacing "try to present" with "present" or "try"

**Section 4**

- Section 4.1: The authors state that the calibration process is repeated 50 times. How do you assure that the you don't get the same result every time? Is the algorithm started with different initial values, or are there stochastic parts in the algorithm?
- P7, L21: add a comma after "algorithms"
- P7, L23: add a comma after "(CMA-ES)"
- P7, L24: I suggest removing "the outstanding"
- P7, L26: remove the parenthesis "(" after "SCE-UA"
- P7, L28: the reference "MA H, et al., 2006) is not present in the bibliography
- P7, L31: add "other" before "three"
- P8, L1 & L2: I assume you mean "normal" instead of "norm"
- P8, L2: I suggest replacing "proven" with "shown"
- P8, L3: replace "on" with "in"
- P8, L4-12 concerning the Bayesian MCMC approach:
  - It appears that the authors used the Metropolis algorithm. If so, please state this.
  - Have these calibration runs been performed specifically for this study or did the authors use the results from Hararuk et al. (2014, 2015)?
  - How is the acceptance probability calculated?
  - How was convergence of the MCMC algorithm diagnosed. What criterion was used?
  - Please provide more information on how the MLE point is determined
  - It is stated that Table 3 provides the detail of the Bayesian MCMC approach. However, other than the number of iterations no information is given
- Figure 4. I assume the box plots show means and spread over the 50 calibration runs. Please indicate this in the caption
- Figure 4: I suggest replacing "exceptions" with "outliers"
- P8 L15: I suggest replacing "measure" with e.g. "applied"
- P8, L16: please revise "As the requirement of Bayesian MCMC…"
- P8, L19: I suggest replacing "On" with "for"
- P8, L20: I suggest removing "only"
- P8, L21: remove "of" before "more"
- P8, L21: consider rephrasing "can exploit better results"
- P8, L22: I suggest replacing "from the aspect" with "with respect to"
- P8, L24: replace "get" with "gets"
- P8, L25: consider rephrasing "promising one"
- P8, L26: consider rephrasing "It is because…"
- P8, L28: I suggest "…matrix and thus that…" with "…matrix, hence…"
- P8, L29: consider rephrasing "extremely critical"
- Figure 5: For all three models, the RBF-SBO algorithm starts out with a considerably lower RMSE at the first iteration, compared to the other algorithms. Please explain this difference. I wonder if the setup of the algorithm somehow gives RBF-SBO an advantage. This would make the comparison unfair.
- P9, L1 & L15: replace "till" with "until"
- P9, L2: add "s" after "simulation"

- P9, L7: consider removing "as we know"
- P9, L7: replace "the" with "an" before "approximation"
- P9, L14: remove "that"
- P9, L14: replace "increasing" with "higher"
- P9, L15: consider rephrasing "keeps ahead"
- P9, L24: consider rephrasing "for the samples"
- P9 L24: "on the other hand" indicates that what follows contradicts what was stated previously. This is not the case here.
- P9, L29: replace "are" with "is" (refers to "performance" not "models")
- P9, L29: consider rephrasing "no one can dominate other two"
- P9, L29: consider rephrasing "all get success"

**Section 5**

- P10, L1: add "state" after "steady"
- P10, L1: replace "Equateion" with "Equation"
- P10, L2: add "the" before "IGBP-DIS"
- P10, L2: I suggest removing "obviously"
- P10, L13: I suggest replacing "sharp" with "narrow"
- P10, L13: I suggest removing "that those are"
- P10, L15: replace "in" with "for"
- P10, L16-17: "On the other hand": see comment for P9 L24, above
- P10, L19: replace "approximate" with "close"
- P10, L19: replace "assigend" with "assigned"
- P10, L20: consider rephrasing "not so reasonable"
- P10, L20: replace "reaches" with "approaches" or "is close to"
- Section 5.2: for CUE_slope and CUE_0 there is considerable mismatch between the mode of the parameter distributions derived by MCMC, and the estimate from the SBO algorithm. But this is only mentioned in passing (P11, L19-21). It needs to be mentioned more explicitly and the potential reasons and consequences should be discussed.
- P11, L5: move "both" from its current location in the sentence to before "CLM-CASA'"
- P11, L14: consider replacing "biomass" with "dynamics" or "processes" since CLM-CASA' also has a microbial biomass pool

**Section 6**

- P11, L23: I don't agree with the statement that "Bayesian MCMC approach has been used to typical SOC models". To my mind most of these models have been tuned either manually or with gradient search algorithms
- P11, L24-25: "owing to approximate one million simulations". The number of required iterations is completely dependent on the specific calibration problem so one cannot state a specific number for calibrating SOC models in general
- P11, L27-28: see comment P2, L30
- P11, L30: I suggest replacing "dominates" with "outperforms"
- P12, L3-4: "it still can find the true parameter values". The mismatch for CUE_slope and CUE_0 in Figure 10 shows that this is not always the case

---

## Author Comment (AC1) · 3 Nov 2017

First of all, the authors would like to thank the reviewer for the valuable comments, suggestions as well as generous recognitions, which would greatly improve the clarity of our presentation and help our revision.

Comment: The quality of the English throughout the manuscript is extremely poor with numerous grammatical errors throughout all the text. Without a great deal of additional editing for language alone this will not be publishable in GMD. All these English language errors, which are far to numerous to call out individually, make it very difficult to undertake a review of scientific merit, but there are a number of areas that clearly require further elaboration and clarification.

Response: Thanks. We have tried our best to conduct several rounds of proofreading and substantially improved English presentation in our revision manuscript.

Comment: Whilst it is certainly challenging, the authors assertion that it is not possible to optimize parameters directly in land surface models such as CLM is not true – see for example Post et al., 2017 JGR-B and reference there in.

Response: We apologize for the confusion. We meant to express the point that optimizing turnover rates and other related parameters with pool-based datasets is computationally too demanding for land surface models. Optimizing flux-related parameters is computationally challenging but possible. The study by Post et al. 2017 was conducted to optimize photosynthesis-related parameters with eddy-flux data. It is a great work. To optimize parameters, such as those we estimated in this study, has to overcome additional computational challenges. For example, it takes a very long spin-up time to run the whole CLM model. Tuning parameters for the whole model requires the extreme computational and temporal cost.

Comment: The assertion that the "structures of land carbon cycle" with ESMs "are almost the same" maybe true but requires evidence and references.

Response: Thanks for your suggestion. Now we cite the paper by Huang et al. 2017, which shows that the matrix equation not only can exactly reproduce the original CLM4.5bgc but also offers the simplicity in coding, diagnostic capacity, and computational efficiency. The latter enables optimizing pool-related parameter estimation.

Comment: It is unclear what are the differences between CLM, CLM-CASA and CLM-CASA C-only. My interpretation is that CLM-CASA C-only is the steady-state approximation detailed in Xia et al, 2012, and the SBO was developed for this. This is important, as the relevance, or otherwise, of this work to informing ESM development can only be understood if the implications of using a surrogate model to parameterize a matrix-based approximation of the steady-state of the simplistic soil component of an old land model are fully articulated. Some additional detail is required here – for example, what are the meteorological drivers, what are the inputs? "NPP" is mentioned, but never explained.

Response: Thanks for your questions. We use the term CLM to refer Community Land Model in a general term. CLM-CASA' is a version CLM3.5 of CLM. The CLM-CASA' C-only version is the same model CLM-CASA' only when we consider the C processes of that model. We developed SBO to optimize parameter estimation for the CLM-CASA' C only version using the steady-state approximation. Parameter optimization cannot easily be done at the non-steady state unless time-series data sets are used as did by Zhou et al. 2013 and 2015.

As the matrix equation results from the re-organization of exact equations as in the original model, the parameter values estimated by SBO can be directly transferred to the original model. Moreover, the matrix representation offers the solution to the land carbon cycle modeling, the National Center for Atmospheric Research (NCAR) land modeling team will adopt the matrix equation as the main frame of CLM in the future version. Thus, any parameter estimation with the matrix equation can be directly used to improve the original model.

We added sentences to clarify this point in our revised manuscript.

Comment: The description of how the specific SBO algorithm and parameter point generation strategies is unclear – what is about the nature of the algorithms chosen that makes them appropriate for this particularly use case? Given the code available in the supplementary material, it is apparent that the various optimization algorithms were implemented in Matlab and relies heavily on material from the File Exchange. Details of this implementation need to be in the main text.

Response: The initial parameter is generated using LHS(Latin Hyper-Cuber Sampling) and we agree that the details of the SBO algorithms should be given. An appendix which includes a detailed description of SBO has been added in the revised manuscript. The reason why SBO outperformed than other global optimization algorithms is that SBO uses a surrogate model to simulate the source model (CLM carbon process) and avoids much bad parameter point ( 'bad' means the high prediction error). By using the surrogate model, the SBO can save many model running times and is appropriate for expensive computation cost models.

Comment: As the authors highlight, "sample size, the nonlinearity and complexity of the real model" all impact surrogate performance. This is partially addressed through the use of three models with different numbers of pools/parameters but not well explained, nor is there reference back to the role of surrogates with ESMs of full complexity.

Response: We agree with the reviewer. To validate this, we carefully choose three different models to evaluate our algorithm. The nonlinearity and complexity of the three models are different. The number of parameters and the equations are different, and the performance of surrogate models are also different. We will add some detailed analysis in the revised version.

Comment: The analysis of the results (Section 5) fails to discuss the implications of the optimizations for CLM-CASA C-only. What does it mean for the model if even when optimized it can only explain 40% of observed variation? Why are so many parameter values right at the edge of their prior range? Are the numbers "biological feasible"? To what extent is the improvement in fit with microbial model due to the inclusion of microbes, or rather due to spatially varying base rates?

Response: We greatly appreciate the reviewer pointing out this issue. It is still not satisfactory to explain 40% of observed variation with the optimized model. This is similar to another study by Hararuk et al. 2014 and much better than the model with default parameter values, which only explains 27% of the observed variation. The unexplained variation is partly due to uncertainty in observations. Indeed, the world homogenized soil data is grid-based map of soil carbon content, which was developed from pedon data. The equation used for the homogenization only can explain 27% of the variation in the original pedon data. That means that the homogenization itself generates 73%

variation. To improve the model-observation fitting, we need to understand uncertainty sources from data, model structure, parameters, and forcing.

The edge-hitting of estimated parameters is usually related to correlations among parameters. We need information of covariances among parameters to resolve the edge-hitting issues.

It is not very clear to us what the reviewer tried to ask with the question Are the numbers "Biological feasible"? Is it about the number of parameters that can be constrained by the dataset we used in this study? In general soil carbon content rich information to constrain several parameters related to soil carbon pool turnover as showed in this study.

The improvement in fit with microbial model is largely due to the nonlinearity, which is more flexible to fit data. It is not clear to us whether the improved fit has anything to do with spatial variation in base rates. We may design a different study to explore this issue.

References: Todd-Brown, K. E. O., J. T. Randerson, W. M. Post, F. M. Hoffman, C. Tarnocai, E. A. G. Schuur, and S. D. Allison (2013), Causes of variation in soil carbon simulations from CMIP5 Earth system models and comparison with observations, Biogeosciences, 10(3), 1717–1736, doi:10.5194/bg-10-1717-2013. Weng, E., and Y. Luo (2011), Relative information contributions of model vs. data to short- and long-term forecasts of forest carbon dynamics, Ecol. Appl., 21(5), 1490–1505, doi:10.1890/09-1394.1. Luo, Y., L. W. White, J. G. Canadell, E. H. DeLucia, D. S. Ellsworth, A. Finzi, J. Lichter, and W. H. Schlesinger (2003), Sustainability of terrestrial carbon sequestration: A case study in Duke Forest with inversion approach, Global Biogeochem. Cycles, 17(1), 1021, doi:10.1029/2002GB001923. Xia, J., Y. Luo, Y. P. Wang, and O. Hararuk (2013), Traceable components of terrestrial carbon storage capacity in biogeochemical models, Global Change Biol., 19, 2104–2116, doi:10.1111/gcb.12172. Zhou, T., and Y. Luo (2008), Spatial patterns of ecosystem carbon residence time and NPP-

driven carbon uptake in the conterminous United States, Global Biogeochem. Cycles, 22, GB3032, doi:10.1029/2007GB002939. Huang YY, XJ Lu, Z Shi, D Lawrence, C Koven, JY Xia, ZG Du, E Kluzek, YQ Luo. (2017). Matrix approach to land carbon cycle modeling: A case study with Community Land Model. Global Change Biology DOI: 10.1111/gcb.13948.

───────────────────────

---

## Author Comment (AC2) · 3 Nov 2017

First of all, the authors would like to thank the reviewer for the detail and valuable comments, suggestions as well as generous recognitions, which would greatly improve the clarity of our presentation and help our revision.

Comment: The fact that the RBF-SBO starts out with a considerably lower RMSE for all three models (Figure 5) suggests that the calibration setup somehow gives RBF-SBO an unfair advantage over the other algorithms. If this is the case, it would have serious consequences for the paper. Possibly the calibrations would have to be redone in a setup that removes this advantage.

Response: The reason that the setup of RBF-SBO is better than other optimization

algorithms is that SBO has a sample step and select the good parameters while other algorithms simply select initial samples randomly. We also repeat the experiments by making the SBO have the same quality setup as other algorithms. The results can be found in the three figs below. The results show that RBS-SBO also has the similar performance on 2-pool carbon model and outperforms other algorithms on CLM-CASA' Carbon Model with the limit of 200 sample runs.

Comment: The description of the methods need to be considerably expanded since much important information is missing, most importantly, on the algorithm itself. Ideally, one should be able to reproduce the approach from the description in the main text, appendix, or supplemental information. However, in this manuscript not nearly enough information is provided for this. For example, I would guess that the algorithm evaluates and rejects several proposal steps using the surrogate model, before a parameter set is deemed good enough to be evaluated by the true model. However, no information is provided as to how these kinds of choices are made. I would suggest including a pseudo-code block to describe the working of the algorithm. Additionally, the surrogate model is constructed based on "radial basis functions" but no additional information is given on how this works. Since the approach for surrogate model is a critical choice (as acknowledged by the authors, P6, L14) this approach needs to be described in much more detail. There are several other places in the text where more information should be provided. These are given in the specific comments below. Parts of these descriptions may be placed in an appendix or online supplement.

Response: Thanks for pointing out the issue. We agree that the details of the algorithms and the "radial basis functions" should be given. The surrogate-based optimization introduction section are refactored in our revised version to make the description more clear and emphasize the main idea of the SBO. The detailed equations and algorithm procedure are given in the Appendix.

Comment: I find the paper a bit biased towards a positive assessment of the algorithm and superiority over other algorithms. The paper would benefit from an additional

discussion section on the possible limitations of the approach, which I'm sure exist. For example, the limitations of using a surrogate model for mimicking complex models is briefly mentioned (P9, L5-11), but its consequences are not further discussed.

Response: The experiments are fair to all presented algorithms including global optimization, MCMC and SBO. The sample size is the same and the results on all three models demonstrate that SBO is better than other parameter calibration methods. The limitation of SBO is that it may be overperformed by global optimization when using more samples. However, the sample size can't be too large for computationally expensive models. Some works such as collaborative tuning are targeted to combine the SBO and global optimization.

Comment: Furthermore, the SBO based estimates strongly disagree with the MCMC estimates for two of the 4-pool microbial model (CUE_slope, and CUE_0; Figure 10). This is briefly mentioned (P11, L19) but not further discussed. Comment: P12, L3-4: "it still can find the true parameter values". The mismatch for CUE_slope and CUE_0 in Figure 10 shows that this is not always the case.

Response: The Figure 10 shows that the some calibrate values of SBO are different from the Bayesian MCMC, and these different values make the prediction error of SBO results is lower than Bayesian MCMC. The parameter values of SBO is better than Bayesian MCMC. According to our understanding, the mismatch of these parameters may be due to the different targets of the parameter selection between SBO and Bayesian MCMC.

Comment: The language in the paper is in general quite poor. There are quite a few spelling and grammar errors, and many sentences are semantically incorrect (e.g. missing or incorrect usage of articles), awkward, or use spoken rather than written English. I've listed a number of them below, but I strongly advise proof-reading by a proficient an editor proficient in the English language. Please check also the citation references, both in the text and in the bibliography. There appear to be quite a few

mistakes.

Response: Many thanks for your valuable suggestions. We have tried our best to conduct several rounds of proofreading and substantially improved English presentation in our revision manuscript.

Comment: From what I can understand from the paper (P3, L5-15) the authors only ran and calibrated soil carbon models, no full land carbon model. Therefore, I find the title somewhat misleading. The approach can probably be used to optimize a full land carbon model, but this has not been shown. I could imagine that the limitations posed by using a surrogate model would become more relevant for a full land carbon model. Hence, I would suggest replacing "land carbon models" with "soil carbon models", or "the soil carbon component of land carbon models".

Response: The title is changed to "soil carbon models" in the revised version

Comment: It is rather unfair to compare computational cost of the SBO approach presented here to that of Bayesian MCMC, since the latter is a sampling algorithm, whereas the former is a optimization algorithm. Sampling schemes are intended to obtain a detailed approximation of the posterior/likelihood function whereas optimization schemes only yield an estimate of the maximum likelihood point. Comparing the computational cost to that of the other optimization approaches would make more sense.

Response: We agree with the reviewer. The Bayesian MCMC is designed to obtain a posterior likelihood function but it can also be used to calibrate parameters to reduce the prediction error of the model. Moreover, we also compare the SBO with known global optimization algorithms.

Comment: Section 2.2: The microbial soil carbon models and the corresponding equations (3)-(16) need to be better explained (e.g. what processes do the different terms in the ODEs represent). For someone not experienced with such models it is currently difficult to understand what's going on.

[Figure]

Response: The detailed introduction of the microbial soil carbon models can be found in the paper "Hararuk, Oleksandra, M. J. Smith, and Y. Luo. Microbial models with data‐driven parameters predict stronger soil carbon responses to climate change." Global Change Biology 21.6 (2015)". I have also added some sentences for clarification in the revised version.

Comment: Section 3: as discussed above the radial basis functions approach needs to be explained, as well as the approach to generate proposal samples

Response: The discussion and introduction are included in the Appendix of the revised version.

Comment: Section 4.1: The authors state that the calibration process is repeated 50 times. How do you assure that the you don't get the same result every time? Is the algorithm started with different initial values, or are there stochastic parts in the algorithm?

Response: Thanks for the question. In fact, even if we start these algorithms (MCMC, global optimization, SBO) with the same initial values, the final calibrated results are different in different running time. It is due to the stochastic nature in these algorithms. We ran each algorithm 50 times and used the average to evaluate the algorithms to eliminate the influence of this kind of randomness.

Comment: P8, L4-12 concerning the Bayesian MCMC approach: -It appears that the authors used the Metropolis algorithm. If so, please state this. -Have these calibration runs been performed specifically for this study or did the authors use the results from Hararuk et al. (2014, 2015)? -How is the acceptance probability calculated? -How was convergence of the MCMC algorithm diagnosed. What criterion was used? -Please provide more information on how the MLE point is determined -It is stated that Table 3 provides the detail of the Bayesian MCMC approach. However, other than the number of iterations no information is given

[Figure]

Response: The Bayesian MCMC (Hararuk et al, 2015, mentioned before) is the Metropolis algorithm. We have got the code from Hararuk and repeated the calibration experiment. This MCMC approach would run 50, 000 samples before ends.

Comment: P11, L23: I don't agree with the statement that "Bayesian MCMC approach has been used to typical SOC models". To my mind most of these models have been tuned either manually or with gradient search algorithms

Response: As far as we know, the bayesian MCMC approach has been used to the two microbial soil carbon models and the carbon cycle component of CLM. Hararuk, Oleksandra, M. J. Smith, and Y. Luo. Microbial models with data‐driven parameters predict stronger soil carbon responses to climate change. Hararuk, Oleksandra, J. Xia, and Y. Luo. Evaluation and improvement of a global land model against soil carbon data using a Bayesian Markov chain Monte Carlo method. Journal of Geophysical Research Biogeosciences 119.3(2014):403-417.

Comment: P11, L24-25: "owing to approximate one million simulations". The number of required iterations is completely dependent on the specific calibration problem so one cannot state aspecific number for calibrating SOC models in general

Response: We agree with the reviewer. It's difficult to estimate the number for different parameter calibration tasks. I have clarified this in our revision version. According to the experiments we conducted in this work, the sample size the SBO requires is less than the Bayesian MCMC method and global optimization algorithms for parameter calibration task.
* * *
[Figure]

Figure 1: 2-pool carbon model

Figure 2: CLM-CASA' Carbon Model

**Fig. 1.**

---

## Author Response (AR1)

This manuscript describes the performance of a surrogate-based approach to calibrating three different soil carbon models relative to three global optimization algorithms and a MCMC algorithm. The results indicate that the surrogate-based optimization employing radial basis functions outperforms the other approaches in nearly all circumstances. Model calibration improves the fit of the models to a global dataset of soil carbon values, with the models incorporating soil microbial dynamics explicitly fitting the data more effectively than a model based on CLM-CASA.

Unfortunately, the quality of the English throughout the manuscript is extremely poor with numerous grammatical errors throughout all the text. Without a great deal of additional editing for language alone this will not be publishable in GMD.

All these English language errors, which are far to numerous to call out individually,

make it very difficult to undertake a review of scientific merit, but there are a number of areas that clearly require further elaboration and clarification.

Whilst it is certainly challenging, the authors assertion that it is not possible to optimize parameters directly in land surface models such as CLM is not true – see for example Post et al., 2017 JGR-B and reference there in.

The assertion that the "structures of land carbon cycle" with ESMs "are almost the same" maybe true but requires evidence and references.

It is unclear what are the differences between CLM, CLM-CASA and CLM-CASA C-only. My interpretation is that CLM-CASA C-only is the steady-state approximation detailed in Xia et al, 2012, and the SBO was developed for this. Some additional detail is required here – for example, what are the meteorological drivers, what are the inputs? "NPP" is mentioned, but never explained. This is important, as the relevance, or otherwise, of this work to informing ESM development can only be understood if the implications of using a surrogate model to parameterize a matrix-based approximation of the steady-state of the simplistic soil component of an old land model are fully articulated.

The description of how the specific SBO algorithm and parameter point generation strategies is unclear – what is about the nature of the algorithms chosen that makes them appropriate for this particularly use case?

Given the code available in the supplementary material, it is apparent that the various optimization algorithms were implemented in Matlab and relies heavily on material from the File Exchange. Details of this implementation need to be in the main text.

As the authors highlight, "sample size, the nonlinearity and complexity of the real model" all impact surrogate performance. This is partially addressed through the use of three models with different numbers of pools/parameters but not well explained, nor is there reference back to the role of surrogates with ESMs of full complexity.

[Figure]

The analysis of the results (Section 5) fails to discuss the implications of the optimizations for CLM-CASA C-only. What does it mean for the model if even when optimized it can only explain 40% of observed variation? Why are so many parameter values right at the edge of their prior range? Are the numbers "biological feasible"? To what extent is the improvement in fit with microbial model due to the inclusion of microbes, or rather due to spatially varying base rates?

Overall, the work described in this manuscript has the potential to inform future land surface model developments, and highlights the possibilities of using surrogate-based optimization at a fraction of the computational cost of MCMC-type approaches. With much improved editing, clarification of the points outlined here, and a more involved discussion of the outcome of optimization exercise – which is the point of the whole exercise after all – hopefully it can be considered more favorably for inclusion in GMD in the future.

[Figure]

Geosci. Model Dev. Discuss.,
https://doi.org/10.5194/gmd-2017-48-RC2, 2017

https://www.geosci-model-dev-discuss.net/gmd-2017-48/gmd-2017-48-RC2-supplement.pdf

[Figure]

**Review of a "Parameter Calibration in Global Land Carbon Models Using Surrogate-based Optimization" by Xu et al.**

Maarten Braakhekke

**General comments**

This manuscript presents a novel approach for calibrating global land carbon cycle models that are computationally costly (i.e. need a long time for a single simulation). The approach, dubbed surrogate-based optimization, uses a uses a computationally cheap surrogate model, which mimics the original model, to generate candidate parameter sets at each iteration. Since the original model is only run for "good" new parameter sets, this approach avoids evaluation of the original model for "bad" parameters, thereby substantially reducing the number of model iterations, and thus computation time. The authors apply the algorithm for the CLM-CASA global land surface model in order to optimize parameters related to soil carbon cycling against global gridded soil carbon stocks from the IGBP-DIS dataset. Additionally, the approach is applied for two other soil carbon models, which explicitly represent microbial dynamics. The calibration results are compared to those of four other optimization schemes and a Bayesian MCMC algorithm.

To my mind the approach is very promising and helps to tackle an important issue with calibrating global models: the high computation cost. I'm not very experienced with optimizing global models and thus I cannot say if there are other techniques that achieve the same thing, and how these compare to the approach presented here. Nevertheless, I think the paper is relevant, and a strong contribution to the field of global modelling. Furthermore, I think the authors were quite thorough in testing the new approach by applying it to three models, and comparing the results to five other optimization/sampling schemes.

However, I do have several criticisms that should be addressed. These relate mainly to the text.

1. The fact that the RBF-SBO starts out with a considerably lower RMSE for all three models (Figure 5) suggests that the calibration setup somehow gives RBF-SBO an unfair advantage over the other algorithms. If this is the case, it would have serious consequences for the paper. Possibly the calibrations would have to be redone in a setup that removes this advantage.

2. The description of the methods need to be considerably expanded since much important information is missing, most importantly, on the algorithm itself. Ideally, one should be able to reproduce the approach from the description in the main text, appendix, or supplemental information. However, in this manuscript not nearly enough information is provided for this. For example, I would guess that the algorithm evaluates and rejects several proposal steps using the surrogate model, before a parameter set is deemed good enough to be evaluated by the true model. However, no information is provided as to how these kinds of choices are made. I would suggest including a pseudo-code block to describe the working of the algorithm. Additionally, the surrogate model is constructed based on "radial basis functions" but no additional information is given on how this works. Since the approach for surrogate model is a critical choice (as acknowledged by the authors, P6, L14) this approach needs to be described in much more detail.

There are several other places in the text where more information should be provided. These are given in the specific comments below. Parts of these descriptions may be placed in an appendix or online supplement.

3. I find the paper a bit biased towards a positive assessment of the algorithm and superiority over other algorithms. The paper would benefit from an additional discussion section on the possible limitations of the approach, which I'm sure exist. For example, the limitations of using a surrogate model for mimicking complex models is briefly mentioned (P9, L5-11), but its consequences are not further discussed. Furthermore, the SBO based estimates strongly disagree with the MCMC estimates for two of the 4-pool microbial model (CUE_slope, and CUE_0; Figure 10). This is briefly mentioned (P11, L19) but not further discussed.

4. The language in the paper is in general quite poor. There are quite a few spelling and grammar errors, and many sentences are semantically incorrect (e.g. missing or incorrect usage of articles), awkward, or use spoken rather than written English. I've listed a number of them below, but I strongly advise proof-reading by a proficient an editor proficient in the English language. Please check also the citation references, both in the text and in the bibliography. There appear to be quite a few mistakes.

5. From what I can understand from the paper (P3, L5-15) the authors only ran and calibrated soil carbon models, no full land carbon model. Therefore, I find the title somewhat misleading. The approach can probably be used to optimize a full land carbon model, but this has not been shown. I could imagine that the limitations posed by using a surrogate model would become more relevant for a full land carbon model. Hence, I would suggest replacing "land carbon models" with "soil carbon models", or "the soil carbon component of land carbon models".

**Specific comments**

**Abstract**

- P1, L11: I suggest to either replace "which can be obviously improved" with "which can obviously be improved", or remove "obviously" altogether

**Section 1**

- P1, L21: "SOC is the largest pool of global land carbon." please provide a reference for this statement.
- P1, L21: I suggest replacing "a famous" with e.g. "the most important".
- P1, L29: I suggest elucidating "agree with". E.g. "For only half of the 11 models the predicted global total SOC falls within estimated range of the HWSD"
- P1, L29: remove "s" in "coefficients"
- P2, L3: remove "the" before "parameter"
- P2, L4: "replace "expensive" with "high"
- P2, L7: remove "the" before "high"
- P2, L8: put "like CLM" between comma's. Also, CLM has not previously been introduced (it is two lines below)
- P2, L15: remove "also"
- P2, L17: replace "the" at the end of the line with "an"
- P2, L18: add "the" before "surrogate"

- P2, L23-24: "Quite a few…benchmark. This sentence is unclear. Consider revising.
- P2, L25: add a comma after "Here"
- P2, L29-30: "On average…Bayesian MCMC". This sentence presents results, and should not be in the introduction. However, I admit this may be a matter of style.
- P2, L30: It is rather unfair to compare computational cost of the SBO approach presented here to that of Bayesian MCMC, since the latter is a *sampling* algorithm, whereas the former is a *optimization* algorithm. Sampling schemes are intended to obtain a detailed approximation of the posterior/likelihood function whereas optimization schemes only yield an estimate of the maximum likelihood point. Comparing the computational cost to that of the other optimization approaches would make more sense.
- P2, L34: Replace "analysis" with "discusses"

**Section 2**
- P3, L1-2: "…their structures of land carbon cycle are almost the same". This is statement is a major oversimplification. I would suggest something like "there are many similarities"
- P3, L6: remove the "s" at the end of "carbons"
- P3, L9: "one of the most popular earth system models in the world". I suggest replacing with "widely used Earth system model"
- P3, L23: add "model" after "CLM-CASA'"
- P3, L24: add "of" after "linear"
- P3, L30: I suggest replacing "The steady solution of equation (1) is solved by Xia et al. (2012):" "the steady state solution of equation is given by (Xia et al. 2012):"
- P4, L1: add a comma after "NPP"
- Section 2.2: The microbial soil carbon models and the corresponding equations (3)-(16) need to be better explained (e.g. what processes do the different terms in the ODEs represent). For someone not experienced with such models it is currently difficult to understand what's going on.
- P4, L21: add "be" after "to"
- P5, L12: add "The" to the start of the sentence and remove the "s" in the second "models"
- Figure 2: Why do gridcells near coastlines have no data?
- P5, L22: replace "gird" with "grid"
- Tables 1 and 2: please provide the units of the parameters

**Section 3**
- Section 3: as discussed above the radial basis functions approach needs to be explained, as well as the approach to generate proposal samples
- P6, L9: I suggest adding "surrogate" before "model"
- P6, L12: I suggest replacing "cancelled" with "avoided"
- P6, L133: I suggest replacing "save much" with e.g. "substantially reduce"
- P6, 24-26: This sentence is rather vague. What is meant with "real variability"?
- P6, L24: please provide a reference for Latin hypercube sampling
- P6, L24: add "which" between "for" and "LHS"
- P7, L3: I suggest replacing "optimum" with "optima"
- P7, L7: I suggest replacing "try to present" with "present" or "try"

**Section 4**

- Section 4.1: The authors state that the calibration process is repeated 50 times. How do you assure that the you don't get the same result every time? Is the algorithm started with different initial values, or are there stochastic parts in the algorithm?
- P7, L21: add a comma after "algorithms"
- P7, L23: add a comma after "(CMA-ES)"
- P7, L24: I suggest removing "the outstanding"
- P7, L26: remove the parenthesis "(" after "SCE-UA"
- P7, L28: the reference "MA H, et al., 2006) is not present in the bibliography
- P7, L31: add "other" before "three"
- P8, L1 & L2: I assume you mean "normal" instead of "norm"
- P8, L2: I suggest replacing "proven" with "shown"
- P8, L3: replace "on" with "in"
- P8, L4-12 concerning the Bayesian MCMC approach:
  - It appears that the authors used the Metropolis algorithm. If so, please state this.
  - Have these calibration runs been performed specifically for this study or did the authors use the results from Hararuk et al. (2014, 2015)?
  - How is the acceptance probability calculated?
  - How was convergence of the MCMC algorithm diagnosed. What criterion was used?
  - Please provide more information on how the MLE point is determined
  - It is stated that Table 3 provides the detail of the Bayesian MCMC approach. However, other than the number of iterations no information is given
- Figure 4. I assume the box plots show means and spread over the 50 calibration runs. Please indicate this in the caption
- Figure 4: I suggest replacing "exceptions" with "outliers"
- P8 L15: I suggest replacing "measure" with e.g. "applied"
- P8, L16: please revise "As the requirement of Bayesian MCMC…"
- P8, L19: I suggest replacing "On" with "for"
- P8, L20: I suggest removing "only"
- P8, L21: remove "of" before "more"
- P8, L21: consider rephrasing "can exploit better results"
- P8, L22: I suggest replacing "from the aspect" with "with respect to"
- P8, L24: replace "get" with "gets"
- P8, L25: consider rephrasing "promising one"
- P8, L26: consider rephrasing "It is because…"
- P8, L28: I suggest "…matrix and thus that…" with "…matrix, hence…"
- P8, L29: consider rephrasing "extremely critical"
- Figure 5: For all three models, the RBF-SBO algorithm starts out with a considerably lower RMSE at the first iteration, compared to the other algorithms. Please explain this difference. I wonder if the setup of the algorithm somehow gives RBF-SBO an advantage. This would make the comparison unfair.
- P9, L1 & L15: replace "till" with "until"
- P9, L2: add "s" after "simulation"

- P9, L7: consider removing "as we know"
- P9, L7: replace "the" with "an" before "approximation"
- P9, L14: remove "that"
- P9, L14: replace "increasing" with "higher"
- P9, L15: consider rephrasing "keeps ahead"
- P9, L24: consider rephrasing "for the samples"
- P9 L24: "on the other hand" indicates that what follows contradicts what was stated previously. This is not the case here.
- P9, L29: replace "are" with "is" (refers to "performance" not "models")
- P9, L29: consider rephrasing "no one can dominate other two"
- P9, L29: consider rephrasing "all get success"

**Section 5**

- P10, L1: add "state" after "steady"
- P10, L1: replace "Equateion" with "Equation"
- P10, L2: add "the" before "IGBP-DIS"
- P10, L2: I suggest removing "obviously"
- P10, L13: I suggest replacing "sharp" with "narrow"
- P10, L13: I suggest removing "that those are"
- P10, L15: replace "in" with "for"
- P10, L16-17: "On the other hand": see comment for P9 L24, above
- P10, L19: replace "approximate" with "close"
- P10, L19: replace "assigend" with "assigned"
- P10, L20: consider rephrasing "not so reasonable"
- P10, L20: replace "reaches" with "approaches" or "is close to"
- Section 5.2: for CUE_slope and CUE_0 there is considerable mismatch between the mode of the parameter distributions derived by MCMC, and the estimate from the SBO algorithm. But this is only mentioned in passing (P11, L19-21). It needs to be mentioned more explicitly and the potential reasons and consequences should be discussed.
- P11, L5: move "both" from its current location in the sentence to before "CLM-CASA'"
- P11, L14: consider replacing "biomass" with "dynamics" or "processes" since CLM-CASA' also has a microbial biomass pool

**Section 6**

- P11, L23: I don't agree with the statement that "Bayesian MCMC approach has been used to typical SOC models". To my mind most of these models have been tuned either manually or with gradient search algorithms
- P11, L24-25: "owing to approximate one million simulations". The number of required iterations is completely dependent on the specific calibration problem so one cannot state a specific number for calibrating SOC models in general
- P11, L27-28: see comment P2, L30
- P11, L30: I suggest replacing "dominates" with "outperforms"
- P12, L3-4: "it still can find the true parameter values". The mismatch for CUE_slope and CUE_0 in Figure 10 shows that this is not always the case

Dear Editor and Referees,

First of all, the authors would like to thank the reviewer for the valuable comments, suggestions as well as generous recognitions, which would greatly improve the clarity of our presentation in the revised version.

**Response to the reviews and the change list in the manuscript**

**Response to Anonymous Referee #1**

**Comment 1.1**: The quality of the English throughout the manuscript is extremely poor with numerous grammatical errors throughout all the text. Without a great deal of additional editing for language alone this will not be publishable in GMD. All these English language errors, which are far to numerous to call out individually, make it very difficult to undertake a review of scientific merit, but there are a number of areas that clearly require further elaboration and clarification.

*Response: Thanks. We have tried our best to conduct several rounds of proofreading and substantially improved English presentation in our revised manuscript.*

**Changes in manuscript**: *We have conducted several rounds of proofreading and substantially improved English presentation (from the beginning to the end of our revised manuscript). We don't mark these changes in the pdf file due to too many modifications.*

**Comment 1.2**: Whilst it is certainly challenging, the authors assertion that it is not possible to optimize parameters directly in land surface models such as CLM is not true – see for example Post et al., 2017 JGR-B and reference there in.

*Response: We apologize for the confusion. We meant to express the point that optimizing turnover rates and other related parameters with pool-based datasets is computationally too demanding for land surface models. Optimizing flux-related parameters is computationally challenging but possible. The study by Post et al. 2017 was conducted to optimize photosynthesis-related parameters with eddy-flux data. It is a great work. To optimize the parameters, such as those we estimated in this study, has to overcome additional computational challenges. For example, it takes a very long spin-up time to run the whole CLM model. Tuning parameters for the whole model requires the extreme computational and temporal cost.*

**Changes in manuscript**: *We clarify this point in Section 1 of the revised manuscript by emphasizing the long spin-up time required for carbon cycle simulation (marked as marked as **Change 1.2**).*

**Comment 1.3**: The assertion that the "structures of land carbon cycle" with ESMs "are almost the same" maybe true but requires evidence and references.

**Response**: *Thanks for your suggestion. Now we cite the paper by Huang et al. 2017, which shows that the matrix equation not only can exactly reproduce the original CLM4.5bgc but also offers the simplicity in coding, diagnostic capacity, and computational efficiency. The latter enables optimizing pool-related parameter estimation.*

**Changes in manuscript**: *We clarify this point in Section 1 (marked as marked as **Change 1.3**).*

**Comment 1.4**: It is unclear what are the differences between CLM, CLM-CASA and CLM-CASA C-only. My interpretation is that CLM-CASA C-only is the steady-state approximation detailed in Xia et al, 2012, and the SBO was developed for this. This is important, as the relevance, or otherwise, of this work to informing ESM development can only be understood if the implications of using a surrogate model to parameterize a matrix-based approximation of the steady-state of the simplistic soil component of an old land model are fully articulated.
Some additional detail is required here – for example, what are the meteorological drivers, what are the inputs? "NPP" is mentioned, but never explained.

**Response**: *Thanks for your questions. We used the term CLM to refer Community Land Model in a general term. CLM-CASA' is a version CLM3.5 of CLM. The CLM-CASA' C-only version is the same model CLM-CASA' only when we consider the C processes of that model. We developed SBO to optimize parameter estimation for the CLM-CASA' C only version using the steady-state approximation. Parameter optimization cannot easily be done at the non-steady state unless time-series data sets are used as did by Zhou et al. 2013 and 2015.*

*As the matrix equation results from the re-organization of exact equations as in the original model, the parameter values estimated by SBO can be directly transferred to the original model. Moreover, the matrix representation offers the solution to the land carbon cycle modeling and the National Center for Atmospheric Research (NCAR) land modeling team will adopt the matrix equation as the main frame of CLM in the future version. Thus, any parameter estimation with the matrix equation can be directly used to improve the original model.*

*We added sentences to clarify this point in our revised manuscript and also to give the detail descriptions of the models.*

**Changes in manuscript**: *We clarify this point in Section 2.1 of our revised manuscript (marked as marked as **Change 1.4**).*

**Comment 1.5**: The description of how the specific SBO algorithm and parameter point generation strategies is unclear – what is about the nature of the algorithms chosen that makes them appropriate for this particularly use case?
Given the code available in the supplementary material, it is apparent that the various optimization algorithms were implemented in Matlab and relies heavily on material from the File Exchange. Details of this implementation need to be in the main text.

**Response**: *The initial parameter is generated using LHS (Latin Hyper-Cube Sampling) and we*

*agree that the details of the SBO algorithms should be given. An appendix which includes a detail description of SBO has been added in the revised manuscript. The reason why SBO outperformed other global optimization algorithms is that SBO uses a surrogate model to simulate the original model (CLM carbon process) and avoids bad parameter points ( 'bad' means the high prediction error). By using the surrogate model, the SBO can save many model running times and is appropriate for expensive computation-cost models.*

**Changes in manuscript**: *We totally refactor Section 3 of our revised manuscript (marked as **Change 1.5-1**). We also point out the advantage of our SBO in Section 4.3.1 of the revised version (marked as **Change 1.5-2**). Moreover, an appendix which includes a detail description of our SBO implementation has been added (marked as **Change 1.5-3**).*

**Comment 1.6**: As the authors highlight, "sample size, the nonlinearity and complexity of the real model" all impact surrogate performance. This is partially addressed through the use of three models with different numbers of pools/parameters but not well explained, nor is there reference back to the role of surrogates with ESMs of full complexity.

**Response**: *We agree with the reviewer. To validate this, we carefully choose three different models to evaluate our algorithm. The nonlinearity and complexity of the three models are different. The number of parameters and the equations are different, and the performance of surrogate models are also different. We will add some detailed analysis in the revised version.*

**Changes in manuscript**: *We clarify this point and point out the value of our SBO for ESMs of full complexity in Section 4.3.1 of the revised version (marked as **Change 1.6**).*

**Comment 1.7**: The analysis of the results (Section 5) fails to discuss the implications of the optimizations for CLM-CASA C-only. What does it mean for the model if even when optimized it can only explain 40% of observed variation? Why are so many parameter values right at the edge of their prior range? Are the numbers "biological feasible"? To what extent is the improvement in fit with microbial model due to the inclusion of microbes, or rather due to spatially varying base rates?

**Response**: *We greatly appreciate the reviewer pointing out this issue. It is still not satisfactory to explain 40% of observed variation with the optimized model. This is similar to another study by Hararuk et al. 2014 and much better than the model with default parameter values, which only explains 33% of the observed variation. The unexplained variation is partly due to uncertainty in observations. Indeed, the world homogenized soil data is grid-based map of soil carbon content, which was developed from pedon data. The equation used for the homogenization only can explain 33% of the variation in the original pedon data. That means that the homogenization itself generates 67% variation. To improve the model-observation fitting, we need to understand uncertainty sources from data, model structure, parameters, and forcing.*

*The edge-hitting of estimated parameters is usually related to correlations among parameters. We need information of covariance among parameters to resolve the edge-hitting issues.*

*It is not very clear to us what the reviewer tried to ask with the question "Are the numbers "Biological feasible?" Is it about the number of parameters that can be constrained by the dataset we used in this study? In general, soil carbon contents rich information to constrain several parameters related to soil carbon pool turnover as showed in this study.*

*The improvement in fit with microbial model is largely due to the nonlinearity, which is more flexible to fit data. It is not clear to us whether the improved fit has anything to do with spatial variation in base rates. We may design a different study to explore this issue.*

**Changes in manuscript**: *We clarify this point on "only explain 42% of observed variation" in Section 5.1 (marked as **Change 1.7-1**). And we also give more explanations on the edge-hitting issue (marked as **Change 1.7-2**).*

**Response to Referee #2**

**Comment 2.1**: The fact that the RBF-SBO starts out with a considerably lower RMSE for all three models (Figure 5) suggests that the calibration setup somehow gives RBF-SBO an unfair advantage over the other algorithms. If this is the case, it would have serious consequences for the paper. Possibly the calibrations would have to be redone in a setup that removes this advantage.

*Response: The reason that the setup of RBF-SBO is better than other optimization algorithms is that SBO has a sample step and selects the good parameters while other algorithms simply select initial samples randomly. We also repeat the experiments by making the SBO have the same quality setup as other algorithms. The results can be found in the figures below. The results show that RBF-SBO has the similar performance on 2-pool microbial model and outperforms other algorithms on 4-pool microbial model and CLM-CASA' Carbon Model with the limit of 200 sample runs.*

[Figure]

*Figure 1: 2-pool microbial model*

[Figure]

*Figure 2: 4-pool microbial model*

[Figure]

*Figure 3: CLM-CASA' Carbon Model*

**Changes in manuscript**: *We clarify the effect of the setup stage in Section 4.3.1 of the revised version (marked as **Change 2.1**).*

**Comment 2.2**: The description of the methods need to be considerably expanded since much important information is missing, most importantly, on the algorithm itself. Ideally, one should be able to reproduce the approach from the description in the main text, appendix, or supplemental information. However, in this manuscript not nearly enough information is provided for this. For example, I would guess that the algorithm evaluates and rejects several proposal steps using the surrogate model, before a parameter set is deemed good enough to be evaluated by the true model. However, no information is provided as to how these kinds of choices are made. I would suggest including a pseudo-code block to describe the working of the algorithm.

Additionally, the surrogate model is constructed based on "radial basis functions" but no additional information is given on how this works. Since the approach for surrogate model is a critical choice (as acknowledged by the authors, P6, L14) this approach needs to be described in much more detail. There are several other places in the text where more information should be provided. These are given in the specific comments below. Parts of these descriptions may be placed in an appendix or online supplement.

*Response*: *Thanks for pointing out the issue. We agree that the details of the algorithms and the "radial basis functions" should be given. The surrogate-based optimization introduction section (Section 3) is refactored in our revised version to make the description more clear and emphasizes the main idea of the SBO. The detailed equations and algorithm procedures are given in the Appendix.*

**Changes in manuscript**: *We totally refactor Section 3 of our revised manuscript (marked as **Change 2.2-1**). Moreover, an appendix which includes a detail description of our SBO implementation has been added (marked as **Change 2.2-2**).*

**Comment 2.3**: I find the paper a bit biased towards a positive assessment of the algorithm and superiority over other algorithms. The paper would benefit from an additional discussion section on the possible limitations of the approach, which I'm sure exist. For example, the limitations of using a surrogate model for mimicking complex models is briefly mentioned (P9, L5-11), but its consequences are not further discussed.

*Response: The experiments are fair to all presented algorithms including global optimization, MCMC and SBO. The sample size is the same and the results on all three models demonstrate that SBO is better than other parameter calibration methods. The reason why SBO outperformed other global optimization algorithms is that SBO uses a surrogate model to simulate the source model (CLM carbon process) and avoids bad parameter points ( 'bad' means the high prediction error). The limitation of SBO is that it may be overperformed by global optimization algorithms when using more samples. However, the sample size can't be too large for computationally expensive models. Some works such as collaborative tuning are targeted to combine the SBO and global optimization.*

**Changes in manuscript**: *We emphasize the reason why our SBO can outperform other global optimization algorithms (marked as **Change 2.3-1**) and point out the potential issue of our SBO (marked as **Change 2.3-2**) in Section 4.3.1 of the revised version.*

**Comment 2.4**: Furthermore, the SBO based estimates strongly disagree with the MCMC estimates for two of the 4-pool microbial model (CUE_slope, and CUE_0; Figure 10). This is briefly mentioned (P11, L19) but not further discussed.
P12, L3-4: "it still can find the true parameter values". The mismatch for CUE_slope and CUE_0 in Figure 10 shows that this is not always the case.

*Response: The Figure 10 shows that some calibrated values of the SBO are different from the Bayesian MCMC, and these different values make the prediction error of SBO results lower than Bayesian MCMC. According to our understanding, the mismatch of these parameters may be due to the different targets of the parameter selection between SBO and Bayesian MCMC.*

**Changes in manuscript**: *We briefly discuss the reason of the mismatch issue for 4-pool microbial model in Section 5.2 (marked as **Change 2.4-1**) and improve the responding statement in Section 6 of the revised version (marked as **Change 2.4-2**)*

**Comment 2.5**: The language in the paper is in general quite poor. There are quite a few spelling and grammar errors, and many sentences are semantically incorrect (e.g. missing or incorrect usage of articles), awkward, or use spoken rather than written English. I've listed a number of them below, but I strongly advise proof-reading by a proficient an editor proficient in the English language. Please check also the citation references, both in the text and in the bibliography. There appear to be quite a few mistakes.

*Response: Many thanks for your valuable suggestions. We have fixed all the grammatical and formatting issues you pointed out, and tried our best to conduct several rounds of proofreading and*

*substantially improved English presentation in our revised manuscript.*

**Changes in manuscript**: ***We have fixed all the grammatical and formatting issues you pointed out.*** *We have also conducted several rounds of proofreading and substantially improved English presentation (**from the beginning to the end of our revised manuscript**). We don't mark these changes in the pdf file due to too many modifications.*

**Comment 2.6**: From what I can understand from the paper (P3, L5-15) the authors only ran and calibrated soil carbon models, no full land carbon model. Therefore, I find the title somewhat misleading. The approach can probably be used to optimize a full land carbon model, but this has not been shown. I could imagine that the limitations posed by using a surrogate model would become more relevant for a full land carbon model. Hence, I would suggest replacing "land carbon models" with "soil carbon models", or "the soil carbon component of land carbon models".

*Response: Thanks. The title has been changed to use "soil carbon models" in the revised version.*

**Changes in manuscript**: *The title has been changed to "Parameter Calibration in Global Soil Carbon Models Using Surrogate-based Optimization" in the revised version (marked as **Change 2.6**).*

**Comment 2.7**: It is rather unfair to compare computational cost of the SBO approach presented here to that of Bayesian MCMC, since the latter is a sampling algorithm, whereas the former is a optimization algorithm. Sampling schemes are intended to obtain a detailed approximation of the posterior/likelihood function whereas optimization schemes only yield an estimate of the maximum likelihood point. Comparing the computational cost to that of the other optimization approaches would make more sense.

*Response: We agree with the reviewer. The Bayesian MCMC is designed to obtain a posterior likelihood function but it can also be used to calibrate parameters to reduce the prediction error. Moreover, we also compare the SBO with known global optimization algorithms in our manuscript.*

**Changes in manuscript**: *We clarify this point in Section 4.2 of the revised version (marked as **Change 2.7**)*

**Comment 2.8**: Section 2.2: The microbial soil carbon models and the corresponding equations (3)-(16) need to be better explained (e.g. what processes do the different terms in the ODEs represent). For someone not experienced with such models it is currently difficult to understand what's going on.

*Response: The detail introduction of the microbial soil carbon models can be found in the paper "Hararuk, Oleksandra, M. J. Smith, and Y. Luo. Microbial models with data-driven parameters predict stronger soil carbon responses to climate change." Global Change Biology 21.6 (2015)". We have added some sentences for clarification in the revised version.*

**Changes in manuscript**: *We clarify this point and cite the references Hararuk, 2014 and 2015 in Section 2.2 of the revised version (marked as **Change 2.8**)*

**Comment 2.9**: Section 3: as discussed above the radial basis functions approach needs to be explained, as well as the approach to generate proposal samples

*Response: The introduction and discussion have been included in the Appendix of the revised version.*

**Changes in manuscript**: *We give more description of radial basis function approach in the appendix (marked as **Change 2.9-1**) and also give more descriptions of the sampling methods in Section 3 of the revised version (marked as **Change 2.9-2**).*

**Comment 2.10**: Section 4.1: The authors state that the calibration process is repeated 50 times. How do you assure that you don't get the same result every time? Is the algorithm started with different initial values, or are there stochastic parts in the algorithm?

*Response: Thanks for the question. In fact, even if we start these algorithms (MCMC, global optimization, SBO) with the same initial values, the final calibrated results are different in different running time. It is due to the stochastic nature of these algorithms. We ran each algorithm 50 times and used the average results for algorithm evaluation to eliminate the influence of this kind of uncertainty.*

**Changes in manuscript**: *We clarify this point in Section 4.1 of the revised version (marked as **Change 2.10**).*

**Comment 2.11**: P8, L4-12 concerning the Bayesian MCMC approach:
-It appears that the authors used the Metropolis algorithm. If so, please state this.
-Have these calibration runs been performed specifically for this study or did the authors use the results from Hararuk et al. (2014, 2015)?
-How is the acceptance probability calculated?
-How was convergence of the MCMC algorithm diagnosed. What criterion was used?
-Please provide more information on how the MLE point is determined
-It is stated that Table 3 provides the detail of the Bayesian MCMC approach. However, other than the number of iterations no information is given

*Response: The Bayesian MCMC (Hararuk et al, 2015, mentioned before) used the Metropolis algorithm. We have got the code from Hararuk and repeated the calibration experiments. This MCMC approach would run 50, 000 samples before ends.*

**Changes in manuscript**: *We clarify this point in Section 4.2 of the revised version (marked as **Change 2.11**).*

**Comment 2.12**: P11, L23: I don't agree with the statement that "Bayesian MCMC approach has

been used to typical SOC models". To my mind most of these models have been tuned either manually or with gradient search algorithms

***Response***: *As far as we know, the Bayesian MCMC approach has been used to the two microbial soil carbon models and the carbon cycle component of CLM.*
*1. Hararuk, Oleksandra, M. J. Smith, and Y. Luo. Microbial models with data-driven parameters predict stronger soil carbon responses to climate change. Global change biology, 2015, 21(6): 2439-2453.*
*2. Hararuk, Oleksandra, J. Xia, and Y. Luo. Evaluation and improvement of a global land model against soil carbon data using a Bayesian Markov chain Monte Carlo method. Journal of Geophysical Research Biogeosciences, 2014, 119(3):403-417.*

**Changes in manuscript**: *We change the statement in Section 6 of the revised version (marked as **Change 2.12**).*

**Comment 2.13**: P11, L24-25: "owing to approximate one million simulations". The number of required iterations is completely dependent on the specific calibration problem so one cannot state a specific number for calibrating SOC models in general

***Response***: *We agree with the reviewer. It's difficult to estimate the number for different parameter calibration tasks. We have clarified this in our revision version. According to the experiments we conducted in this work, to achieve the same optimization accuracy, the sample size the SBO requires is less than the Bayesian MCMC method and global optimization algorithms for parameter calibration task.*

**Changes in manuscript**: *We change the statement in Section 6 of the revised version (marked as **Change 2.13**).*

[revised manuscript text omitted]

---

## Referee Report (RR1)

**Review of "Parameter Calibration in Global Soil Carbon Models Using Surrogate-based Optimization" (version 2)**

The authors have made numerous improvements. I also appreciate the clear marking of changes in the PDF which allows cross-referencing with replies to comments. However, for a number of criticisms I am unsatisfied with the reply and/or modifications to the manuscript. I also have a few new comments.

**Replies to specific comments**

**Comment 2.1:** The fact that the RBF-SBO starts out with a considerably lower RMSE for all three models (Figure 5) suggests that the calibration setup somehow gives RBF-SBO an unfair advantage over the other algorithms. If this is the case, it would have serious consequences for the paper. Possibly the calibrations would have to be redone in a setup that removes this advantage.

**Response:** The reason that the setup of RBF-SBO is better than other optimization algorithms is that SBO has a sample step and selects the good parameters while other algorithms simply select initial samples randomly. We also repeat the experiments by making the SBO have the same quality setup as other algorithms. The results can be found in the figures below. The results show that RBFSBO has the similar performance on 2-pool microbial model and outperforms other algorithms on 4-pool microbial model and CLM-CASA' Carbon Model with the limit of 200 sample runs.

*If I understand the reply correctly, the left most point in the graphs of Figs 5a-c for the SBO algorithm corresponds to the first iteration of the loop indicated by steps 4-6 in the pseudo-code block depicted on p 9—not to the initial parameter sets S0 (derived in step 1 of the pseudo-code block). I suppose the reason for this is the fact that S0 represents multiple parameter sets needed to construct the initial surrogate model, so one cannot pick a parameter set that is truly the first. It is good that the authors show that also with "the same quality setup as the other algorithms" SBO shows good performance. Nevertheless, in my view, Fig 5 is misleading, because the "real" model has already been run a number of times before the start of the graph. For the other algorithms on the other hand, (I assume) the left-most point of the graph is truly the first time the model is run. For a more honest comparison I would suggest to shifting the curve of SBO to the right by adding the number of initial simulations to the x-axis value; e.g. if there are 50 initial simulations, the curve would shift 50 points to the right. The same can be done for the other algorithms in case they also require a number of initial simulations. I also find the changes made but the authors (change 2.1; marked sentence starting on P11L28) insufficient. In my view this sentence does not explain the apparent advantage of SBO in Fig 5 to the average reader.*

**Comment 2.4:** Furthermore, the SBO based estimates strongly disagree with the MCMC estimates for two of the 4-pool microbial model (CUE_slope, and CUE_0; Figure 10). This is briefly mentioned (P11, L19) but not further discussed. P12, L3-4: "it still can find the true parameter values". The mismatch for CUE_slope and CUE_0 in Figure 10 shows that this is not always the case.

**Response:** The Figure 10 shows that some calibrated values of the SBO are different from the Bayesian MCMC, and these different values make the prediction error of SBO results lower than Bayesian MCMC. According to our understanding, the mismatch of these parameters may be due to the different targets of the parameter selection between SBO and Bayesian MCMC.

**Changes in manuscript:** We briefly discuss the reason of the mismatch issue for 4-pool microbial model in Section 5.2 (marked as Change 2.4-1) and improve the responding statement in Section 6 of the revised version (marked as Change 2.4-2)

*The added/modified section marked with change 2.4-1 (starting P14L26) is unclear. What is meant with "different targets"? If the same cost/objective function is used, the mode of the MCMC derived distribution should match with optimal value derived with the optimization algorithm. Given that the SBO derived value for CUE0 has negligible posterior probability according to the MCMC derived distribution (Fig 10), while it has a lower RMSE than that of the parameter set corresponding to the mode, suggests that something has gone wrong with the MCMC sampling. Most likely the MCMC algorithm has gotten stuck in a local minimum located close to the global minimum.*

**Comment 2.5:** The language in the paper is in general quite poor. There are quite a few spelling and grammar errors, and many sentences are semantically incorrect (e.g. missing or incorrect usage of articles), awkward, or use spoken rather than written English. I've listed a number of them below, but I strongly advise proof-reading by a proficient an editor proficient in the English language. Please check also the citation references, both in the text and in the bibliography. There appear to be quite a few mistakes.

**Response:** Many thanks for your valuable suggestions. We have fixed all the grammatical and formatting issues you pointed out, and tried our best to conduct several rounds of proofreading and substantially improved English presentation in our revised manuscript.

*The English has improved but it is not yet at a sufficient level in my view. There are still quite a few awkward sentences, and some grammatically incorrect ones. Particularly the use of articles and punctuation should be improved.*

**Comment 2.7:** It is rather unfair to compare computational cost of the SBO approach presented here to that of Bayesian MCMC, since the latter is a sampling algorithm, whereas the former is a optimization algorithm. Sampling schemes are intended to obtain a detailed approximation of the posterior/likelihood function whereas optimization schemes only yield an estimate of the maximum likelihood point. Comparing the computational cost to that of the other optimization approaches would make more sense.

**Response:** We agree with the reviewer. The Bayesian MCMC is designed to obtain a posterior likelihood function but it can also be used to calibrate parameters to reduce the prediction error. Moreover, we also compare the SBO with known global optimization algorithms in our manuscript.

*From the reply it seems that the authors do not agree with me, as opposed to their first sentence. I stand by my point that optimization algorithms and sampling algorithms have different purposes and the latter are not a sensible choice when one is calibrating a computationally expensive model. The authors show this themselves because the CLM-CASA model requires 1,000,000 simulations to get convergence (P11L3), while the optimization algorithms (also non-SBO) get a similar result in less than 1000. However, when discussing performance, the authors are still comparing their optimization algorithm to the*

*sampling* algorithm MCMC. For example, in the last sentence of the abstract "Meanwhile, the corresponding computation cost required is only one thousandth of that with Bayesian MCMC.", and in the conclusions (point 2). This is misleading. Readers not familiar with MCMC will think that the authors have made an incredible performance improvement, which this not the case. Please limit comparing performances to that of the other optimization algorithms (PSO, SCE-UA, CMA-ES), or at least make it clear that Bayesian MCMC has a different purpose.

**Comment 2.11:** P8, L4-12 concerning the Bayesian MCMC approach:

- It appears that the authors used the Metropolis algorithm. If so, please state this.
- Have these calibration runs been performed specifically for this study or did the authors use the results from Hararuk et al. (2014, 2015)?
- How is the acceptance probability calculated?
- How was convergence of the MCMC algorithm diagnosed. What criterion was used?
- Please provide more information on how the MLE point is determined
- It is stated that Table 3 provides the detail of the Bayesian MCMC approach. However, other than the number of iterations no information is given

**Response:** The Bayesian MCMC (Hararuk et al, 2015, mentioned before) used the Metropolis algorithm. We have got the code from Hararuk and repeated the calibration experiments. This MCMC approach would run 50, 000 samples before ends.

**Changes in manuscript:** We clarify this point in Section 4.2 of the revised version (marked as Change 2.11).

*I have difficulty understanding the paragraph related to this comment (change 2.7; starting from P10L25):*

- *What is exactly constitutes the proposal step and the moving step?*
- *According to the Metropolis rule the acceptance probability is determined by the ratio of the unnormalized posterior density for the proposal parameter set and that of the current parameter set. Is this what is meant with "a probability of acceptance determined by prediction error is calculated"*
- *Also "The final calibrated parameter set is estimated by Maximum likelihood estimator (MLE) with an accepted parameter chain." Is this an additional calculation at every iteration or done after the calibration?*

**Additional questions/comments**

- Briefly inspecting the supplemental information, I found that the authors used the Matlab "Surrogate Model Optimization Toolbox", implemented by Julliane Mueller. This is a substantial amount of work that has been done by others. The method description, however, makes no mention of this—initially I was under the impression that the authors implemented the SBO algorithm themselves. The authors refer to two publications of Mueller et al. in the text but only

for general descriptions of SBO. Please make clear in the main text that you used the implementation of Mueller et al. and refer to the appropriate papers.

- P8L25: it is not clear what is meant with "exploration" and "exploitation"
- P9L2: Sentence starting with "It is worthy noted…". Not clear what is meant here.
- P11L8: Sentence starting with "For each algorithm". Not sure what is meant here, That the algorithm is truncated after 100 simulations? If so, why is this done--to limit computation time? Wouldn't the results be strongly dependent on the initial parameter set(s)?
- Figures 7a/b: the quality of the graphs is really low. I would suggest to use a version with a higher resolution
- Figure 2: why do the cells near the cost have no data?
- Table 1: what is meant with "exit rate"? Turnover/decomposition rate? If so I would suggest to use these terms

---

## Author Response (AR2)

Dear Editor Christoph Müller,

We greatly appreciate your granting us another opportunity to revise the manuscript. We are also thankful to the reviewers for their valuable comments and suggestions, which are extremely helpful for us to improve the manuscript. As a consequence, the manuscript has been substantially improved. We hope you will find our revision is satisfactory.

Below please find our responses to comments by the two reviews and the change list in the manuscript

Sincerely,

Wei Xue, Ph.D.
Associate Professor
Department of Computer Science and Technology and Department of Earth System Science, Tsinghua University
Beijing, China, 100084

**Letter of Response**

Note that comments from referees are in italics and our responses in normal shape.

**Referee #1**

*This paper tests the performance of surrogate modeling approach for estimating parameters in three soil carbon models relative to selection of commonly used direct optimization approaches and an MCMC approach. The results suggest their surrogate modeling approach is amongst the most effective and is the most efficient (by their measure at least).*

**Response**: No response needed

**Comment 1.1:**
*The description of the surrogate modeling and optimization approaches are relatively complete, but details about the models themselves, and how they have been used, are incomplete and confusing. This distracts from the main aspects of the work detailing the optimizations. This confusion is not helped by the authors repeatedly making claims about working with Earth system models and global land surface models, when in reality they working with a matrix approximation of a sub-component of such complex models, or models that are not part of such systems at all. Whilst it is exciting to see such approaches being applied in this way, the usability and applicability with fully functionally ESMs must not be overstated.*

*It is important to have clarity here, as one of the main arguments you make is how beneficial surrogate models are when working with computationally expensive, highly complex Earth system models. Yet, in reality the surrogate model is being built for the matrix approximation, which in itself was developed to address many of the issues related to parameterization of ESMs. Often in the manuscript you described the days to weeks of wall clock time associated with MCMC model runs in comparison to the efficiency of the surrogate. But is this really the case with the matrix model? Similarly, the 2- and 4-pool models are not highly complex (relative to an ESM), and it is unclear if there are real world benefits from developing a surrogate for such models (that is to say, is the efficiency gain really worth the expense in developing the surrogate for a model of this complexity?)*

**Response**: We greatly appreciate the comment. We clarified this point and made changes in several places of the manuscript. For example, we have revised the second and third paragraphs in the introduction (lines 7-24 on page 2) to clarify those three models for parameter calibration as: (marked as **Change 1.1-1**)

"Considering the high similarity in the structures of the 11 ESMs, the difference in

the SOC simulations mainly results from parameterizations (Todd Brown et al., 2013). Thus, parameter calibration is among the top priorities to improve prediction of global land carbon cycle dynamics (Luo et al., 2016). However, the parameter calibration with global observations has not been widely applied, owing to the high computational cost. A matrix approach has been recently developed to reorganize the carbon balance equations in the original ESMs into one matrix equation without changing any modeled C cycle processes and mechanisms (Luo et al. 2003, 2017; Huang et al. 2018). The matrix land carbon cycle models can be semi-analytically solved to obtain steady-state solutions faster than the original models by tens and hundreds times (Xia et al. 2013). As a consequence, the matrix approach makes parameter estimation and calibration possible. The matrix approach has been successfully used for the parameter calibration to constrain SOC turnover and microbial process with Bayesian Markov chain Monte Carlo (MCMC) algorithm (Harauk et al., 2014, 2015; Shi et al. 2018).

Bayesian MCMC is a sampling-based approach and usually requires a large number of simulations for building an acceptable parameter chain. For instance, over 500,000 simulations are required during the parameter calibration of soil carbon models (Xu et al. 2006). Even using high-performance computers, complex land models, like the latest version of Community Land Model (CLM5.0), require a very long spin-up time for carbon cycle simulation, leading to several hours or days for one simulation. Although the matrix approach has been developed to enable data assimilation of global land carbon cycle models (Harauk et al., 2014 and 2015, Shi et al. 2018), Bayesian MCMC computationally is still very expensive for calibrating global land models. More effective and efficient parameter calibration algorithms are urgently needed."

We have also revised the second last paragraph in the Introduction (lines 9-17 on page 3) to clarify the purpose of evaluating SBO with the three models as: (marked as **Change 1.1-2**)

"Most studies on both global and surrogate optimizations focus on the mathematical function benchmarks like Comparing Continuous Optimizers, abbreviated as COCO (Hansen et al., 2010; Wang and Duan, 2014). However, the optimization of the mathematical functions may be extremely different from the parameter calibration of complex real-world models. In this paper, we explore the state-of-the-art surrogate optimization method for parameter calibration of three SOC models: CLM coupled with Carnegie-Ames-Stanford Approach biogeochemistry model (CLM-CASA') and two microbial models, as used in studies by Hararuk et al. (2014, 2015). Although the three models are computationally attainable for parameter calibration, we compare the performance of surrogate-based optimization to advanced global optimization algorithms and the data assimilation method to examine the potential of SBO. The SBO may be extended to other complex global land models."

We also added one sentence in the Conclusion (Lines 16-17, page 16) to clarify the usefulness of this study as: (marked as **Change 1.1-3**)

"Although the three SOC models used in this analysis are not computationally unattainable for parameter calibration, what we have learned about SBO from this study can be potentially applied to more complex models."

Moreover, we have thoroughly revised section 2 to clearly describe the three SOC models.

**Comment 1.2:**
*It is unclear how the three soil C models were actually run. What are the inputs? What is the climate forcing? Where do they come from? Are they a time series, or mean values used to estimate a steady state solution? Rather than describing the model equations – which could be moved to appendices – a more holistic description of the modeling approach including details of domain, forcing, NPP inputs would be beneficial.*
*Related to this is confusion as to how the CLM-CASA' C-only model is being used. Section 2.1 talks about the matrix approximation in general terms – but not specifically about its use in this case. Whilst acknowledging there are references describing this, some detail is required directly in this manuscript.*

**Response:** We apologize for the confusion of model description. We have thoroughly revised section "2 Global Land Carbon Models, Data, and Cost Function" on pages 3-6 to clarify the descriptions of the three SOC models (marked as **Change 1.2**). The inputs are defined by NPP and the outputs are the carbon content of each pool at a steady state. Hope the reviewers will find our revision satisfactory.

**Comment 1.3:**
*The results themselves are encouraging, and what might be expected, with the exception of the relatively poor performance of the MCMC scheme in the case of the 2- and 4-pool models. The authors suggest this "may be mainly due to the different targets of the parameter selection". This needs further elaboration because I don't really understand what that means, and it really is a surprising result.*

**Response**: Same as the response for Comment 2.2.

**Comment 1.4:**
*Figure 2 needs some clarification. Why is the data so "blocky" and covers such a limited extent of the land mass? Is that the domain over which the model was actually run?*

**Response**: Thanks for pointing out this problem. We have replotted Figure 2 (marked as **Change 1.4**). For the previous version, we followed the method using in Hararuk et al. 2014.

Hararuk O, Xia J, Luo Y. Evaluation and improvement of a global land model against

soil carbon data using a Bayesian Markov chain Monte Carlo method. Journal of Geophysical Research: Biogeosciences, 2014, 119(3): 403-417.

**Comment 1.5:**
*It seems like much of the appendix replicates Section 3.3 – but is actually written better and is clearer. I would suggest merging them into Section 3.3 alone.*

**Response**: Thanks for the valuable suggestion. We have merged the appendix into Section 3.3 (marked as **Change 1.5**).

**Comment 1.6:**
*The quality of the English is very variable across different sections of the manuscript. The abstract and introduction are very poor with numerous language errors – far too many for this reviewer to list individually and any resubmission will require a thorough proof reading.*

**Response**: Thanks. We have carefully edited the manuscript and improved English presentation in our revised manuscript, especially for the abstract and introduction sections.

**Referee #2**

**Comment 2.1:**
*If I understand the reply correctly, the left most point in the graphs of Figs 5a-c for the SBO algorithm corresponds to the first iteration of the loop indicated by steps 4-6 in the pseudo-code block depicted on p 9—not to the initial parameter sets S0 (derived in step 1 of the pseudo-code block). I suppose the reason for this is the fact that S0 represents multiple parameter sets needed to construct the initial surrogate model, so one cannot pick a parameter set that is truly the first. It is good that the authors show that also with "the same quality setup as the other algorithms" SBO shows good performance. Nevertheless, in my view, Fig 5 is misleading, because the "real" model has already been run a number of times before the start of the graph. For the other algorithms on the other hand, (I assume) the left-most point of the graph is truly the first time the model is run. For a more honest comparison I would suggest to shifting the curve of SBO to the right by adding the number of initial simulations to the x-axis value; e.g. if there are 50 initial simulations, the curve would shift 50 points to the right. The same can be done for the other algorithms in case they also require a number of initial simulations. I also find the changes made but the authors (change 2.1; marked sentence starting on P11L28) insufficient. In my view this sentence does not explain the apparent advantage of SBO in Fig 5 to the average reader.*

**Response**: Thanks. We shift the curves of SBO to the right by adding the number of initial simulations along with the x-axis (marked as **Change 2.1-1**). In our work, we

used Latin Hypercube Sampling (LHS) method and sampled 18 initial points to start our SBO algorithms.

Moreover, we have also revised the third paragraph of the Section 4.3.1 (lines 6-10 on page 12) to clarify the advantages of our SBO as: (marked as **Change 2.1-2**)

"Compared to the global optimization algorithms, SBO has two main advantages. First, SBO samples an initial parameter set to build a surrogate model, and the building process is a learning process which can better understand the parameter space, thus help conduct better optimization. Second, SBO can avoid some bad parameter points ('bad' means high prediction error), which are not supposed to be evaluated with the help of the surrogate model."

**Comment 2.2:**
*The added/modified section marked with change 2.4-1 (starting P14L26) is unclear. What is meant with "different targets"? If the same cost/objective function is used, the mode of the MCMC derived distribution should match with optimal value derived with the optimization algorithm. Given that the SBO derived value for CUE0 has negligible posterior probability according to the MCMC derived distribution (Fig 10), while it has a lower RMSE than that of the parameter set corresponding to the mode, suggests that something has gone wrong with the MCMC sampling. Most likely the MCMC algorithm has gotten stuck in a local minimum located close to the global minimum.*

**Response**: Thanks. This is an interesting argument for our experiments of both SBO and MCMC. We carefully re-analyzed both designs of experiments and try our best to explain the reason of mismatch of results between our SBO and MCMC.

First, the objective function of optimization algorithms (such as DE, PSO or our SBO) is averaged RMSE between the results of model and IGBP-DIS observation data in our case. The target of optimization algorithms is to minimize the averaged RMSE.

Different from optimization algorithms, the Bayesian MCMC approach do not have an optimization objective function or loss function. MCMC is an adaptive sampling method and we use maximum likelihood estimator (MLE) to estimate the parameters from the samples generated by MCMC. MCMC use an acceptance probability to determine whether accept a new parameter point or not. The probability of acceptance can be summarized as follows (Marshall et al., 2004):

$$\text{P}(c^{k-1}|c^{new}) = \min\{1, \frac{P(O|c^{new})P(c^{new})}{P(O|c^{k-1})P(c^{k-1})}\}$$

$P(O|c)$ is a likelihood function of parameter c; $P(c)$ is prior probability density function of parameter c; Assuming that the prediction errors are normally distributed and uncorrelated, hence, the $P(O|c)$ can be calculated as follows (Hararuk et al., 2014):

$$P(O|c) \propto \exp\{-\sum_{i=1}^{N} \frac{(X_i - O_i)^2}{2\sigma_i^2}\}$$

Where $O_i$ is soil C reported by IGBP-DIS at ith grid cell, $X_i$ is soil C simulated by CLM-CASA' or microbial models at ith grid cell; $\sigma_i^2$ is the variance of a measurement at a grid cell. From the two equations above, the MCMC tends to keep the parameter points, whose $\sum_{i=1}^{N} \frac{(X_i - O_i)^2}{2\sigma_i^2}$ (named as r') are lower, which is different from the objective function or cost function of optimization (averaged RMSE presented in Eq.17 in our manuscript):

$$r = \sqrt{\frac{1}{N} \sum_{i=1}^{N} (X_i - O_i)^2}$$

Owing to the difference between r and r', the final estimation of MCMC could be a little bit different from the solution of parameter calibration with optimization algorithms in our case.

**Comment 2.3:**
*The English has improved but it is not yet at a sufficient level in my view. There are still quite a few awkward sentences, and some grammatically incorrect ones. Particularly the use of articles and punctuation should be improved.*

**Response**: Thanks. We have carefully edited the manuscript and improved English presentation in our revised manuscript.

**Comment 2.4:**
*From the reply it seems that the authors do not agree with me, as opposed to their first sentence. I stand by my point that optimization algorithms and sampling algorithms have different purposes and the latter are not a sensible choice when one is calibrating a computationally expensive model. The authors show this themselves because the CLM-CASA model requires 1,000,000 simulations to get convergence (P11L3), while the optimization algorithms (also non-SBO) get a similar result in less than 1000. However, when discussing performance, the authors are still comparing their optimization algorithm to the sampling algorithm MCMC. For example, in the last sentence of the abstract "Meanwhile, the corresponding computation cost required is only one thousandth of that with Bayesian MCMC.", and in the conclusions (point 2). This is misleading. Readers not familiar with MCMC will think that the authors have made an incredible performance improvement, which this not the case. Please limit comparing performances to that of the other optimization algorithms (PSO, SCE-UA, CMA-ES), or at least make it clear that Bayesian MCMC has a different purpose.*

**Response**: Thanks for pointing out this problem again. We agree with the reviewer and

have modified the corresponding sentences in our revised version to limit the performance comparison.

**Comment 2.5:**
*I have difficulty understanding the paragraph related to this comment (change 2.7; starting from P10L25):*
*• What is exactly constitutes the proposal step and the moving step?*
*• According to the Metropolis rule the acceptance probability is determined by the ratio of the unnormalized posterior density for the proposal parameter set and that of the current parameter set. Is this what is meant with "a probability of acceptance determined by prediction error is calculated"*
*• Also "The final calibrated parameter set is estimated by Maximum likelihood estimator (MLE) with an accepted parameter chain." Is this an additional calculation at every iteration or done after the calibration?*

**Response**: Thanks. We have refactored the Section 4.2 and given the detail description of the MCMC for better clarification (marked as **Change 2.5**).

The proposal step generates a new parameter set $p^{k+1}$ from the previously accepted parameter set $p^k$ through a proposal distribution $q(p^{k+1}|p^k)$. The proposal distribution is obtained using 50,000 simulations sampled following a uniform distribution. In the moving step, the probability of acceptance is calculated and used to determine whether accept the new parameter set $p^{k+1}$. And then the accepted one is added to the final sample set.

As mentioned in the Response of Comment 2.4, the MCMC method tends to keep the parameter points, whose $\sum_{i=1}^{N} \frac{(X_i - O_i)^2}{2\sigma_i^2}$ are lower. We use MLE estimator to run once on the final sample set generated by the AM algorithm to estimate the best parameter values, that are approximate to the minimum of $\sum_{i=1}^{N} \frac{(X_i - O_i)^2}{2\sigma_i^2}$.

**Comment 2.6:**
*Briefly inspecting the supplemental information, I found that the authors used the Matlab "Surrogate Model Optimization Toolbox", implemented by Julliane Mueller. This is a substantial amount of work that has been done by others. The method description, however, makes no mention of this—initially I was under the impression that the authors implemented the SBO algorithm themselves. The authors refer to two publications of Mueller et al. in the text but only for general descriptions of SBO. Please make clear in the main text that you used the implementation of Mueller et al. and refer to the appropriate papers.*

**Response**: Thanks for pointing out this issue We have added a new reference of the surrogate model optimization toolbox.

**Comment 2.7:**

*P8L25: it is not clear what is meant with "exploration" and "exploitation"*

**Response**: In our work, "exploration" refers to searching in an unfamiliar area of the parameter space to learn about it and avoid trapping into some local optimum, and "exploitation" means fast convergence in some area. Balancing both the exploration and the exploitation makes our SBO can find global optimum and do not waste more model runs on the meaningless parameter areas.

**Comment 2.8:**

*P9L2: Sentence starting with "It is worthy noted…". Not clear what is meant here.*
*P11L8: Sentence starting with "For each algorithm". Not sure what is meant here, That the algorithm is truncated after 100 simulations? If so, why is this done--to limit computation time? Wouldn't the results be strongly dependent on the initial parameter set(s)?*
*Figures 7a/b: the quality of the graphs is really low. I would suggest to use a version with a higher resolution*

**Response**: We have improved the presentation for our manuscript, and provided the high quality graphs of Figure 7.

**Comment 2.9:**

*Figure 2: why do the cells near the cost have no data?*

**Response**: Same as the response for Comment 1.4.

**Comment 2.10:**

*Table 1: what is meant with "exit rate"? Turnover/decomposition rate? If so I would suggest to use these terms*

**Response**: Yes. We have replaced "exit rate" with "decomposition rate" in the revised version.

[revised manuscript text omitted]